# Intestine-specific removal of DAF-2 nearly doubles lifespan in *Caenorhabditis elegans* with little fitness cost

Yan-Ping Zhang[1,2,4], Wen-Hong Zhang [1,2,4], Pan Zhang[1], Qi Li [3], Yue Sun[1], Jia-Wen Wang[1], Shaobing O. Zhang[3], Tao Cai [1], Cheng Zhan [1] & Meng-Qiu Dong [1,2]✉

Twenty-nine years following the breakthrough discovery that a single-gene mutation of *daf-2* doubles *Caenorhabditis elegans* lifespan, it remains unclear where this insulin/IGF-1 receptor gene is expressed and where it acts to regulate ageing. Using knock-in fluorescent reporters, we determined that *daf-2* and its downstream transcription factor *daf-16* are expressed ubiquitously. Using tissue-specific targeted protein degradation, we determined that intracellular DAF-2-to-DAF-16 signaling in the intestine plays a major role in lifespan regulation, while that in the hypodermis, neurons, and germline plays a minor role. Notably, intestine-specific loss of DAF-2 activates DAF-16 in and outside the intestine, causes almost no adverse effects on development and reproduction, and extends lifespan by 94% in a way that partly requires non-intestinal DAF-16. Consistent with intestine supplying nutrients to the entire body, evidence from this and other studies suggests that altered metabolism, particularly down-regulation of protein and RNA synthesis, mediates longevity by reduction of insulin/IGF-1 signaling.

One of the breakthrough discoveries in biology from the last 30 years is the finding that ancient genetic pathways control animal lifespan[1–3]. The first identified and extensively validated one is the insulin/insulin-like growth factor 1 (IGF-1) signaling (IIS) pathway[4–6]. Reduction of the IIS extends the lifespan in *Caenorhabditis elegans*[4,7,8], *Drosophila*[9,10], and mice[11,12]. Moreover, single nucleotide polymorphisms of IIS component genes are tightly linked to human longevity[13–15]. Reducing IIS also alleviates pathologies of age-associated diseases in animal models, including those for Alzheimer's and Parkinson's[16,17]. These studies indicate that the IIS pathway is a promising target for developing anti-ageing therapeutics. However, applying this knowledge in practice has not been possible because of the many essential functions of IIS, including glucose metabolism, lipid metabolism, growth, and reproduction[18–21].

In humans and other mammals, insulin and IGF peptides are synthesized in and secreted primarily from the pancreatic β-cells and the liver, respectively. They are carried by the bloodstream to target tissues where they bind to and activate their cell-surface receptors. In humans, both the insulin receptor (IR) and the IGF-1 receptor (IGF-1R) are expressed in nearly all tissues (https://portal.brain-map.org/). In mammals, studies of IIS mostly focus on β-cells, liver, muscle, and adipose tissue, where IIS is crucial in maintaining homeostasis of glucose metabolism and energy metabolism[18,22]. Given these essential functions, mutations of IRs are linked to several inheritable genetic diseases including type A insulin resistance syndrome, Donohue syndrome, and Rabson-Mendenhall syndrome[23]. In healthy people, sensitivity to insulin declines with age, accompanied by an increasing probability of developing serious chronic conditions such as type 2

[1]National Institute of Biological Sciences, Beijing, Beijing, China. [2]Beijing Key Laboratory of the Cell Biology of Animal Aging, Beijing, China. [3]Laboratory of Metabolic Genetics, College of Life Sciences, Capital Normal University, Beijing, China. [4]These authors contributed equally: Yan-Ping Zhang, Wen-Hong Zhang. ✉e-mail: dongmengqiu@nibs.ac.cn

diabetes and obesity[24–26]. As such, it was surprising when reduction of IIS was discovered to extend lifespan of *C. elegans*[4]. Interestingly, some forms of general IIS reduction are accompanied by a longevity phenotype in humans, mice, and dogs, although they also exhibit an undesirable growth retardation phenotype[1].

The *daf-2* gene encodes the sole *C. elegans* homolog of IR/IGF-1R[27]. Other core components of IIS include AGE-1/PI3-K, PDK-1, AKT-1/2, and DAF-16/FoxO[27–31]. The IIS kinase cascade, from DAF-2 to AKT-1/2, maintains a relatively short wild-type (WT) lifespan by inhibiting the transcription factor (TF) DAF-16. Loss-of-function (*lf*) mutations of the upstream kinases all produce a remarkable longevity phenotype in a *daf-16* dependent manner. For example, the canonical *daf-2(e1370)* allele doubles *C. elegans* lifespan, and the lifespan extension is completely abolished by deletion of *daf-16*[4]. Reduction of *C. elegans* IIS from young adulthood produces a stronger longevity phenotype than reducing it in later periods[32]. In addition to a long lifespan, *C. elegans* IIS mutants are highly pleiotropic, exhibiting varying degrees of developmental defects, reduced brood size, and increased fat storage[27,33]. Similar phenotypes are found in IIS mutants of mice[34–36].

Various methods, from genetic mosaic analysis[37] to tissue-specific RNAi[38] or transgene rescue[39], have been tried to identify where IIS acts to control ageing, development, or metabolism in *C. elegans*. Special emphasis has been placed on whether IIS regulates ageing from a single tissue or not. Despite the efforts, no consensus has been reached: experimental data have suggested the nervous system[40], the intestine[39], or multiple cell lineages[37] as the sites of lifespan regulation by IIS.

The main source of the confusion is that the exact expression pattern of *daf-2* is unclear. The *daf-2* gene is 50 kb long, with large introns, multiple transcription start sites and alternative splicing sites; there are also multiple, long cDNAs. These characteristics make it difficult to reliably determine the expression pattern of *daf-2* with traditional transgene approaches. Immunostaining and in situ hybridization did not produce consistent results, either, with one showing DAF-2 in XXX cells and neurons[41] and the other showing DAF-2 in the germline (http://nematode.nig.ac.jp/db2/index.php). Cell- or tissue-specific transcriptomic studies suggest that *daf-2* is widely expressed[42–44], but it needs confirmation using orthogonal methods as omics data come with a certain level of false identifications. In contrast to the lack of consensus on *daf-2* expression, independent studies using fluorescent transgene reporters all agree that *daf-16* is ubiquitously expressed in somatic tissues[45–48].

Using the CRISPR/Cas9 genome editing technology and a tissue-specific targeted protein degradation system, we determined that DAF-2 and DAF-16 are both expressed ubiquitously in the somatic and reproductive tissues, and that both regulate ageing of the entire body from the intestine. Degradation of DAF-2 in the intestine activates not only DAF-16 in the same tissue, but also DAF-16 in other tissues, and cross-tissue signaling from DAF-2 to DAF-16 contributes to longevity. Furthermore, degradation of intestinal DAF-2 nearly doubles *C. elegans* lifespan with little or no effect on development and reproduction. In other words, two major pleiotropic phenotypes caused by reducing IIS throughout the body can be uncoupled from the longevity phenotype by manipulating IIS in a tissue-specific manner.

This study clarifies the confusion regarding the question of which tissue initiates the longevity signal in IIS mutants[38–40]. We find that a commonly used neuronal promoter has leaky expression in the intestine, which explains why some "neuron-specific" inactivation of DAF-2 extends lifespan substantially. Our finding that both DAF-2 and DAF-16 act primarily in the intestine to regulate lifespan strengthens an early finding that *daf-16* activity is required in the intestine–not neurons–to support *daf-2* longevity[39]. As the primary function of the intestine is to digest food and to absorb nutrients, initiation of the longevity signal in IIS mutants possibly involves alteration in the supplying of nutrients by the intestine to the entire body, with the consequence of inducing organism-wide metabolic changes. In support of this, whole-worm and tissue-specific RNA-seq analyses show that longevity by loss of DAF-2 is associated most prominently with down-regulation of protein and RNA metabolism, including synthesis and degradation of both types of biomolecules. The clarification that it is the intestine that initiates the main longevity signal in the *daf-2* mutant highlights a commonality between lifespan extension by IIS reduction and that by dietary restriction[49,50]: nutrient supply and nutrient sensing.

## Results

### DAF-2/IR/IGF-1R is widely expressed from embryos to adults
To ensure accurate detection of the endogenous expression pattern of *daf-2*, we designed two detection strategies, one focusing on the protein, the other on the mRNA (Fig. 1a). To visualize the DAF-2 protein, we knocked in the coding sequence (CDS) of mNeonGreen–a green fluorescent protein 3–5 times as bright as the green fluorescent protein (GFP)[51,52]–immediately after the last amino acid codon of the *daf-2* gene on chromosome III (Fig. 1a) using the CRISPR/Cas9 genome editing technique. Phenotypic assays showed that this mNeonGreen tag did not perturb the function of the DAF-2 protein to which it was attached (Supplementary Fig. 1). DAF-2::mNeonGreen was detected in neurons, XXX cells, vulval cells, germ cells, and oocytes (Fig. 1b). In the last three types of cells, DAF-2::mNeonGreen had clear plasma membrane (PM) localization as expected for a cell surface receptor. In the neurons and XXX cells (Fig. 1b), a pair of specialized hypodermal cells of neural endocrine function, strong DAF-2::mNeonGreen signals were seen in the cell bodies and along processes. Presence of DAF-2 in vulval cells was not suggested before, but Nakdimon et al. showed that *daf-2(lf)* suppressed the multivulva phenotype induced by hyperactivation of RAS/MAPK signaling[53]. Therefore, the above results suggest a cell-autonomous regulatory function of IIS in vulval development. The DAF-2::mNeonGreen-expressing neurons included all of the ciliated sensory neurons marked by the *osm-6* promoter-driven mCherry protein (Supplementary Fig. 2a).

Because DAF-2 is localized on the PM, it may be difficult to detect using the mNeonGreen tag in cells with large surface area where the signal may be too thin. Therefore, we took an alternative approach to visualize the cells expressing *daf-2* mRNA. Downstream of the *daf-2* CDS, we knocked in the sequence of an intercistronic region from the *C. elegans* SL2-type operon, followed by a nuclear localization sequence (NLS), the CDS of GFP, the CDS of mNeonGreen, and another NLS, which we referred to as the Nuclear Ultrabright GFP::mNeonGreen Fluorescent protein (NuGFP) cassette (Fig. 1a). In this approach, expression of NuGFP is tied to that of *daf-2* from the endogenous locus, but after trans-splicing, the NuGFP protein is synthesized independently of DAF-2. This high-sensitivity *daf-2* expression reporter was readily detectable in most *C. elegans* cells, including the ones that had been missed by the DAF-2::mNeonGreen fusion protein marker, that is, the intestine, hypodermis, gonadal sheath, and body wall muscles (BWM) (Fig. 1c). With NuGFP, expression of *daf-2* was observed starting in 2-cell embryos, and the expression continued throughout development and adulthood (Fig. 1d and Supplementary Fig. 2b–g).

### DAF-16/FoxO is widely expressed from embryos to adults
To visualize the pattern of endogenous expression of *daf-16*, we knocked in the CDS of GFP right before the stop codon of the *daf-16* CDS on chromosome I using the CRISPR/Cas9 genome editing technique (Fig. 2a). Other than illuminating the endogenously expressed DAF-16 proteins, the GFP tag did not interfere with DAF-16 function (Supplementary Fig. 1).

In unstressed animals, such as those kept at standard culture conditions (15–20 °C, well-fed), DAF-16::GFP was dispersed throughout the cell. To better recognize cells expressing DAF-16::GFP, we induced nuclear translocation of DAF-16::GFP by placing the worms on a glass

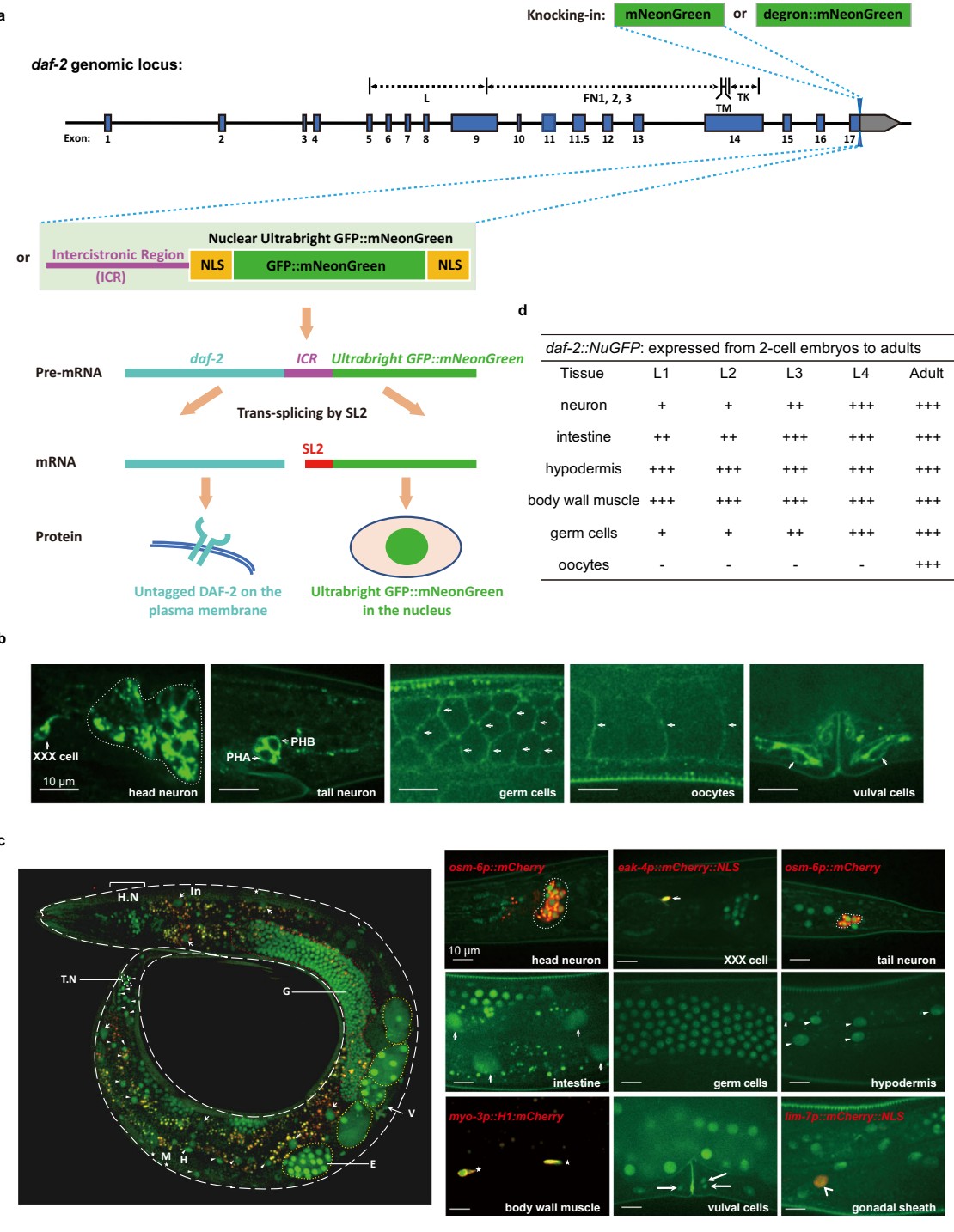

**Fig. 1 | Endogenous expression of the *daf-2* gene detected in most or all *C. elegans* cells from embryos to adults. a** Schematic of two strategies to characterize the endogenous expression pattern of *daf-2*. The coding sequence of the mNeonGreen, mNeonGreen::degron (top panel), or NuGFP cassette (bottom panel) is knocked into the *daf-2* genomic locus before the stop codon by CRISPR/Cas9 genome editing, which allows detection of *daf-2* expression at the protein level or at the mRNA level, respectively. Blue boxes, coding regions; line, non-coding regions; gray boxes, 3′ untranslated regions. **b** Expression pattern of *daf-2* at the protein level indicated by the DAF-2::mNeonGreen at day 1 of adulthood. A similar pattern of expression was observed in four independent experiments. **c** Expression pattern of *daf-2* at the mRNA level indicated by the NuGFP reporter (NLS$^{SV40}$::GFP::mNeonGreen::NLS$^{EGL-13}$) at day 1 of adulthood. Left panel, overview of

the expression of *daf-2* mRNA throughout the whole body. H.N, head neuron; G, germ line, indicated by the circled red dotted line; In, intestine, indicated by short arrows; V, vulval cells, indicated by long arrow; E, embryo, indicated by the circled gray dashed line; H, hypodermal cells, indicated by triangles; T.N, tail neuron, indicated by the circled white dotted line; M, body wall muscle, indicated by asterisks. Right panel, local expression patterns of *daf-2* mRNA. *osm-6p::mCherry*, ciliated sensory neuron marker; *eak-4p::mCherry::NLS$^{EGL-13}$*, XXX-cell-specific marker; *myo-3p::H1::mCherry*, body-wall-muscle-specific marker; *lim-7p::mCherry::NLS$^{EGL-13}$*, gonadal-sheath-specific marker. A similar pattern of expression was observed in two independent experiments. **d** Summary of the spatiotemporal expression pattern of NuGFP reporter. +, expression intensity of NuGFP; -, expression of NuGFP is not detectable.

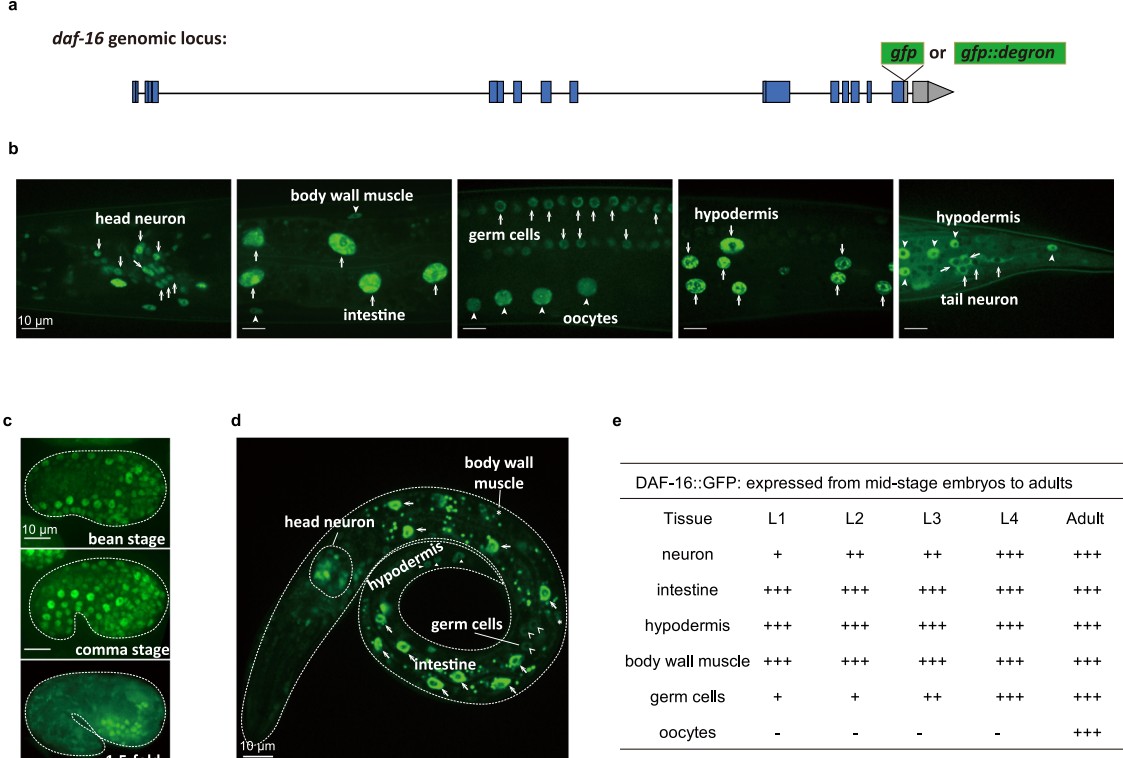

**Fig. 2 | Ubiquitous presence of DAF-16 detected in the *C. elegans* soma and germline. a** Schematic of knocking in the coding sequence of GFP or GFP::degron into the *daf-16* genomic locus before the stop codon using CRISPR/Cas9 genome editing. Blue boxes, coding regions; line, non-coding regions; gray boxes, 3′ untranslated regions. Expression patterns of DAF-16::GFP at day 1 of adulthood (**b**), embryonic stage (**c**), and L1 larval stage (**d**). A similar pattern of expression was observed in two independent experiments. **e** Summary of the spatiotemporal expression patterns of DAF-16::GFP. +, expression intensity of DAF-16::GFP; -, expression of DAF-16::GFP is not detectable.

slide atop an agarose cushion for about 5 min before epifluorescence imaging. We found that DAF-16::GFP was expressed ubiquitously in most or all somatic tissues, such as neurons, intestine, BWM, and hypodermis, and also in the germ cells and oocytes (Fig. 2b). Germline expression of DAF-16::GFP was not detected by earlier transgene reporters[45–48].

Temporally, ubiquitous expression of DAF-16::GFP from the endogenous locus was detected from the embryonic bean stage to adulthood (Fig. 2c–e and Supplementary Fig. 3).

## DAF-2 controls lifespan from the intestine

To settle the controversy regarding the site of action of insulin/IGF-1 signaling in lifespan regulation, we adopted the auxin-induced protein degradation (AID) system[54] (Supplementary Fig. 4a). This system allowed us to achieve tissue-specific DAF-2 or DAF-16 degradation in the WT or the long-lived *daf-2(e1370)* mutant background, respectively.

Using CRISPR/Cas9 technology, we generated knock-in (KI) strains respectively expressing DAF-2::degron::mNeonGreen (Fig. 1a) or DAF-16::GFP::degron (Fig. 2a) from the endogenous *daf-2* or *daf-16* locus. The double tag of a degron sequence and a fluorescent protein sequence enables facile detection of the expression of the fusion protein and auxin-induced target protein degradation. Next, to the existing single-copy insertion (SCI) strains expressing TIR-1 in all cells (*ieSi57*), intestinal cells (*ieSi61*), or germ cells (*ieSi38*)[54], we added neuron-, hypodermis-, BWM-, gonadal-sheath-, and XXX-cell-specific TIR-1 expressing lines by replacing the promoter sequence of *eft-3* or *sun-1* with that of *rgef-1*, *dpy-7*, *myo-3*, *lim-7*, or *eak-4* (Supplementary Fig. 4b). Before using these five promoters, we verified their tissue specificity by expressing NuGFP or mCherry under their control

(Supplementary Fig. 4c–g). A total of eight TIR-1 expressing chromosomes II or IV were each combined with the DAF-2::degron::mNeonGreen or DAF-16::GFP::degron chromosome through genetic crossing. Auxin-induced tissue-specific degradation of fluorescently labeled DAF-2 or DAF-16 was verified (Supplementary Figs. 5 and 6). We also verified that exposure to 1 mM auxin had no effect on the lifespan of WT worms with or without SCI expression of TIR-1 (*ieSi57*) (Supplementary Fig. 7). Additionally, auxin-induced degradation sustained in old worms, as demonstrated using the neuron-specific DAF-2 AID strain (Supplementary Fig. 8).

Next, we examined the effect of tissue-specific degradation of DAF-2 on lifespan (Fig. 3 and Supplementary Data 1). We found that degrading neuronal DAF-2 increased WT lifespan by 18.6% (Fig. 3a), much less than what would be expected from previous *daf-2(lf)* rescue experiments using the tissue-specific-promoter-driven transgene arrays[40]. Degrading DAF-2 in the germline or the hypodermis respectively increased lifespan by 6.4% and 13.7% (Fig. 3b and c), whereas degrading DAF-2 in the BWM, gonadal sheath, or XXX cells had no effect on lifespan (Fig. 3d–f). In contrast, degrading DAF-2 in the intestine extended the *C. elegans* lifespan by 94.4% (Fig. 3g). These results showed unambiguously that intestinal DAF-2 is essential in lifespan regulation, while neuronal, hypodermal, and germline DAF-2 each play a minor role.

Of note, the worms in which DAF-2 was degraded throughout the body had a lifespan that was 266.5% of the WT (Fig. 3h), outliving the canonical hypomorphic *daf-2(e1370)* mutant and the intestinal DAF-2 AID worms, whose lifespans were 206.0% and 193.3% of the WT, respectively (Fig. 3h). Here, FUDR was used to prevent the strong egg-laying defective (Egl) phenotype of whole-body DAF-2 AID from interfering with the lifespan assay. This Egl phenotype, possibly having

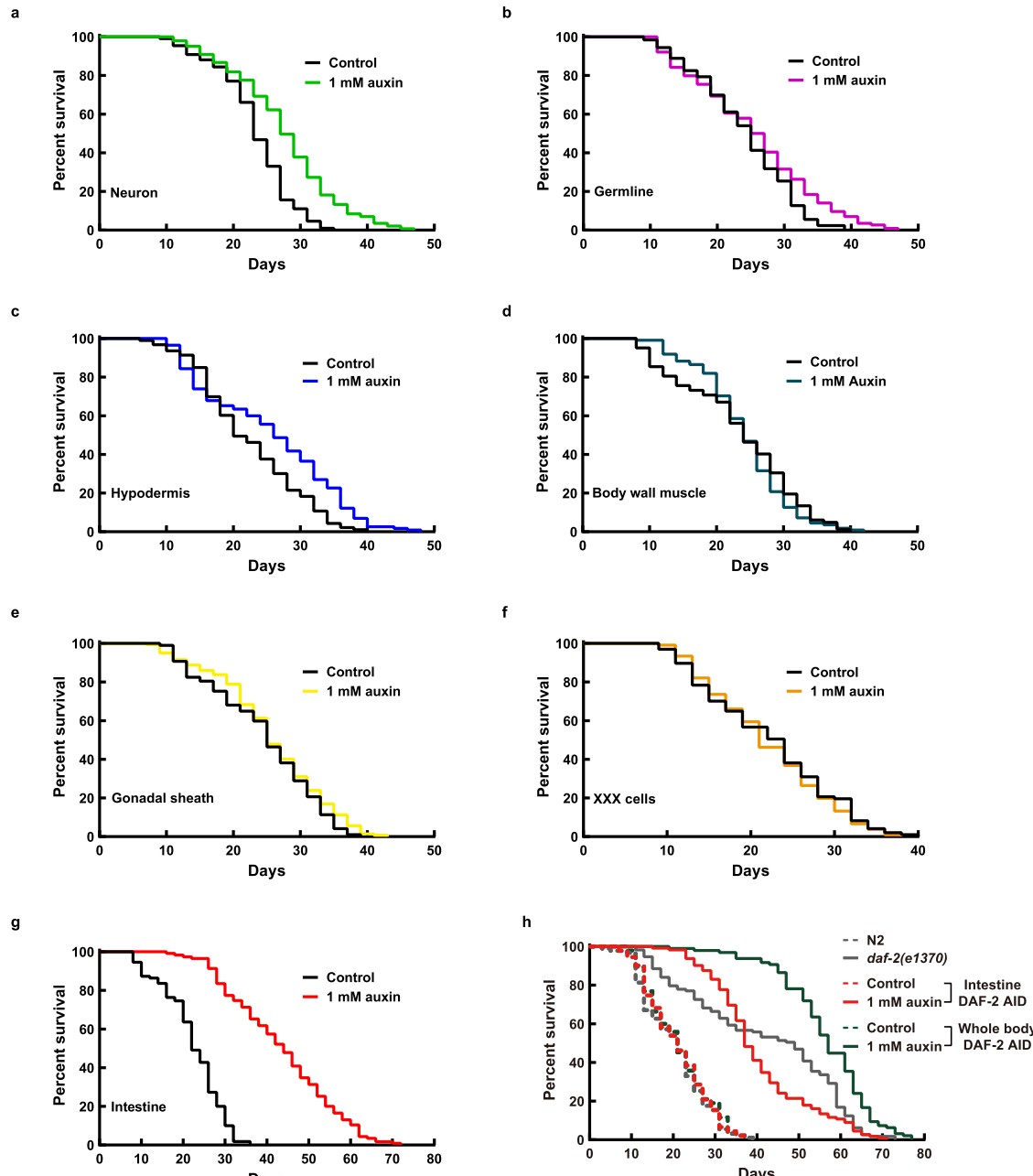

**Fig. 3 | Intestine-specific degradation of DAF-2 extended *C. elegans* lifespan by 94.4%.** Degrading DAF-2 in the neurons (**a**), germline (**b**), and hypodermis (**c**) increased the lifespan by 18.6% ($p < 0.0001$), 6.4% ($p = 0.014$), and 13.7% ($p = 0.001$), respectively. **d–f** Degrading DAF-2 in the body wall muscle (**d**), gonadal sheath (**e**), or XXX cells (**f**) had no effect on WT lifespan ($p > 0.05$). **g** Degrading DAF-2 in the intestine increased WT lifespan by 94.4% ($p < 0.0001$). **h** Degrading DAF-2 in the whole body increased WT lifespan by 166.5% (green solid line versus green dashed line, $p < 0.0001$), which outlived the canonical hypomophic *daf-2(e1370)* mutant worms and the intestinal DAF-2 AID worms (gray solid line and red solid line, respectively). *p* values are calculated by log-rank tests. See survival statistics in Supplementary Data 1. Source data are provided as a Source Data file.

to do with the vulval expression of *daf-2*, causes internal hatching of eggs. Degrading DAF-2 in the intestine or the other tissues examined (Fig. 3a–g) caused no obvious Egl phenotype.

The above results demonstrate that for IIS-mediated lifespan regulation, although DAF-2 in the intestine plays the most important role, DAF-2 activity in other tissues is required for the longevity phenotype to the fullest extent (whole-body DAF-2 AID 166.5% > intestine 94.4% + neuron 18.6% + hypodermis 13.7% + germline 6.4% + others 0%). This is a reminder that ageing is a systemic phenomenon involving cooperation among different tissues.

Previous studies have shown that the *daf-2* mutant worms have enhanced innate immunity and are resistant to bacterial infection,

which contributes to their longevity[55–57]. In this study, we found that worms lacking intestinal DAF-2 lived a long life on bacteria killed or not by gentamicin, about 94% longer than WT worms on live bacteria or 28% longer than WT worms on killed bacteria (Supplementary Fig. 9b). Cox proportional hazard regression analysis revealed that gentamicin reduced the hazard ratio by 0.24, while degradation of intestinal DAF-2 by 0.71 (Supplementary Fig. 9d). The reduction of the hazard ratio by degradation of intestinal DAF-2 is comparable to that by *daf-2(e1370)* or by degradation of DAF-2 throughout the body. These results indicate that longevity by intestinal DAF-2 AID is more than enhancing innate immunity to reduce death by bacterial infection through the intestine.

## Mutation of *daf-2* extends lifespan via intestinal DAF-16

The longevity phenotype of *daf-2(e1370)* is fully dependent on *daf-16*[4], therefore one obvious question is in which tissue DAF-16 mediates this effect. Previously, Libina et al. examined three tissues and found that re-introducing *daf-16* from a transgene to the intestine, but not muscles or neurons, partially restored longevity in the *daf-2; daf-16* double mutant background[39]. However, using tissue-specific RNAi, Uno et al. found that both the neuronal and the intestinal *daf-16* activities are important for lifespan extension by reduction of IIS[38]. Here, we examined the role of DAF-16 in seven tissues in mediating the *daf-2* longevity (Fig. 4 and Supplementary Data 1). We found that auxin-induced degradation of DAF-16 in the neurons, germline, or hypodermis shortened the *daf-2(e1370)* lifespan by no more than 15.6% (Fig. 4a–c), while that in BWM, gonadal sheath, or XXX cells had no effect (Fig. 4d–f). In comparison, degradation of intestinal DAF-16 shortened the *daf-2 (e1370)* lifespan by 40.1% (Fig. 4g), which means that intestinal DAF-16 mediated 72.2% of the lifespan extension by *daf-2(e1370)*. Degrading DAF-16 throughout the body shortened the *daf-2(e1370)* lifespan by 57.8%, making it slightly shorter than the lifespan of WT animals, which was 44.4% of the *daf-2(e1370)* lifespan (Fig. 4h). The effects of tissue-specific DAF-16 degradation resonated those of tissue-specific DAF-2 degradation on WT lifespan (Fig. 3), and clearly indicate that the intestine is the single most important tissue from which IIS regulates lifespan.

Nuclear accumulation is a sign of DAF-16 activation, due to alleviation of the inhibitory phosphorylation by AKT-1/2 on DAF-16 following inactivation of AKT kinase or the upstream kinases AGE-1 and DAF-2[27–31]. In adult animals, we found that endogenously expressed DAF-16::GFP from a KI allele (Fig. 2) accumulated only in the intestine following degradation of DAF-2 in the same tissue (Fig. 4i). Moreover, 83.2% of the extra lifespan gained by degrading intestinal DAF-2 required intestinal DAF-16 (Fig. 4j). Taken together, these results demonstrate that DAF-2 action in the intestine itself is controlling the localization and activation of DAF-16, and intracellular signaling from DAF-2 to DAF-16 mediates most of the effect of intestinal DAF-2 on lifespan.

## Worms lacking intestinal DAF-2 develop and reproduce well

Having established that the intestine is the major site for lifespan regulation by IIS, we asked whether lifespan extension may occur without causing the other pleiotropic phenotypes associated with previously studied *daf-2* mutations[33]. We found that intestine-specific degradation of DAF-2, which nearly doubled lifespan (Figs. 3g and 4g) greatly alleviated the development impairment associated with *daf-2* and caused no reproductive defects (Fig. 5).

The canonical *daf-2(e1370)* mutant worms form dauers at -0.5% penetrance at 20 °C, but this increases to 100% at 25 °C, which is the upper temperature limit for standard *C. elegans* culture[33]. Here we found that worms lacking DAF-2 throughout the worm body formed dauers at 100% at 25 °C, 20 °C, and even 15 °C. In contrast, worms lacking DAF-2 only in the intestine did not form dauers at 25 °C (Fig. 5a). When the culture temperature was increased to 27 °C, they formed transient dauers: >90% remained as dauers for no more than a few days before they resumed reproductive development (Supplementary Fig. 10). This bears comparison with *daf-2* RNAi, which extends *C. elegans* lifespan and causes dauer formation at 45% penetrance at 27 °C[32] but not at all at 25 °C or lower[32,58,59]. The similar effects of *daf-2* RNAi and intestinal DAF-2 AID may be explained by the intestine's sensitivity to RNAi (see discussion).

The developmental rate of the *daf-2(e1370)* mutant is also slower than WT. A total of 91.9% of freshly laid WT eggs developed into adults after 64 h at 20 °C, while 81.3% of the *daf-2(e1370)* population were still at the L3 stage. With 93.7% of the population reaching L4 or adulthood under the same conditions, the intestinal DAF-2 AID worms developed faster than the *daf-2(e1370)* worms and slightly slower than WT (Fig. 5b).

With respect to brood size, *daf-2(1370)* animals laid an average of 198 eggs per worm at 20 °C, which is 26% fewer than WT. In contrast, there was no difference in the number of eggs laid per worm between the intestinal DAF-2 AID worms and control worms (Fig. 5c).

For the lipid storage phenotype, we found that degrading intestinal DAF-2 elevated the triacylglycerol (TAG) content by 2.4-fold relative to the control animals, which recapitulated the metabolic phenotype of the *daf-2(e1370)* mutant (Fig. 5d). Among all the pleiotropic phenotypes examined, this lipid storage phenotype is the only one that remains associated with the long-lived intestinal DAF-2 AID worms. In short, degrading DAF-2 only in the intestine uncouples the longevity phenotype from the developmental and reproductive defects associated with a general reduction of IIS.

## Loss of intestinal DAF-2 lowers RNA and protein metabolism

The many pleiotropic phenotypes of the *daf-2* mutants make it difficult to identify gene expression changes that are associated specifically with longevity. A study of 75 publicly available microarray datasets has identified 1663 genes that are up-regulated (Class I targets) and 1733 genes that are downregulated (Class II targets) in *daf-2(-)* worms compared to the *daf-2(-); daf-16(-)* control[60]. These differentially expressed genes (DEGs) include both direct and indirect targets of DAF-16. Our next-generation RNA-seq analysis of *daf-2(e1370)* versus WT (N2) worms uncovered a total of 2463 DEGs ($p$. adjust < 0.05), of which 1509 were upregulated and 953 were downregulated in *daf-2(e1370)* worms (Fig. 6a, left panel, and Supplementary Data 2). Contrasting the KEGG pathways enriched from the Class I or Class II targets (Fig. 6a, middle panel) and those from the RNA-seq data following gene set enrichment analysis (GSEA) (Fig. 6a, right panel), it is evident that RNA-seq data revealed extensive downregulation of multiple pathways related to protein and RNA metabolism, in both synthesis and degradation. Decreased protein turnover in long-lived *daf-2* worms has been found repeatedly in previous proteomics studies and was shown to contribute positively to the longevity of *daf-2* worms[61–63]. Here, we show that this information is in the transcriptome data, discernable by GSEA (Fig. 6a, right panel).

Since intestine-specific DAF-2 degradation produced a strong longevity phenotype with significantly reduced side effects, we reasoned that it could help separate gene expression changes associated with longevity from those associated with the developmental or reproductive phenotypes of *daf-2* worms. We thus conducted RNA-seq analysis of worms subjected to tissue-specific degradation of DAF-2. The RNA-seq data of three biological replicates (Supplementary Fig. 11 and Supplementary Data 3) showed that the gene expression changes induced by degrading DAF-2 in the intestine and degrading DAF-2 in the whole body are closer to each other than either is to the other treatments (Fig. 6b and Supplementary Fig. 12). For both, the upregulated genes were enriched in fatty acid degradation and peroxisome, and the downregulated genes were enriched in pathways related to protein metabolism, RNA metabolism, and DNA repair.

Notably, RNA-seq analysis underscored downregulation of genes functioning in protein and RNA metabolism, which includes translation, transcription, protein degradation, and RNA degradation, as common features shared by *daf-2(e1370)*, whole-body, and intestine-specific DAF-2 AID (Fig. 6a, b). Previous studies have shown that decreasing protein synthesis extends lifespan[62,64–67] and accounts for ~40% of the lifespan extension by *daf-2(e1370)*[68]. In comparison, little is known about how down-regulation of RNA metabolism contributes to *daf-2* longevity.

Since ribosomal RNAs constitute the majority of total cellular RNAs, we quantified the rRNAs using qRT-PCR. We found that in the *daf-2* mutant, 5S, 18S, and 26S rRNAs all decreased to less than 26% of the WT level, while ITS1 and ITS2, representing the precursor rRNA (pre-rRNA), also decreased to 52% and 34%, respectively (Fig. 6c). Further, knocking down either *M28.5/snu-13* or *fib-1*, two RNA

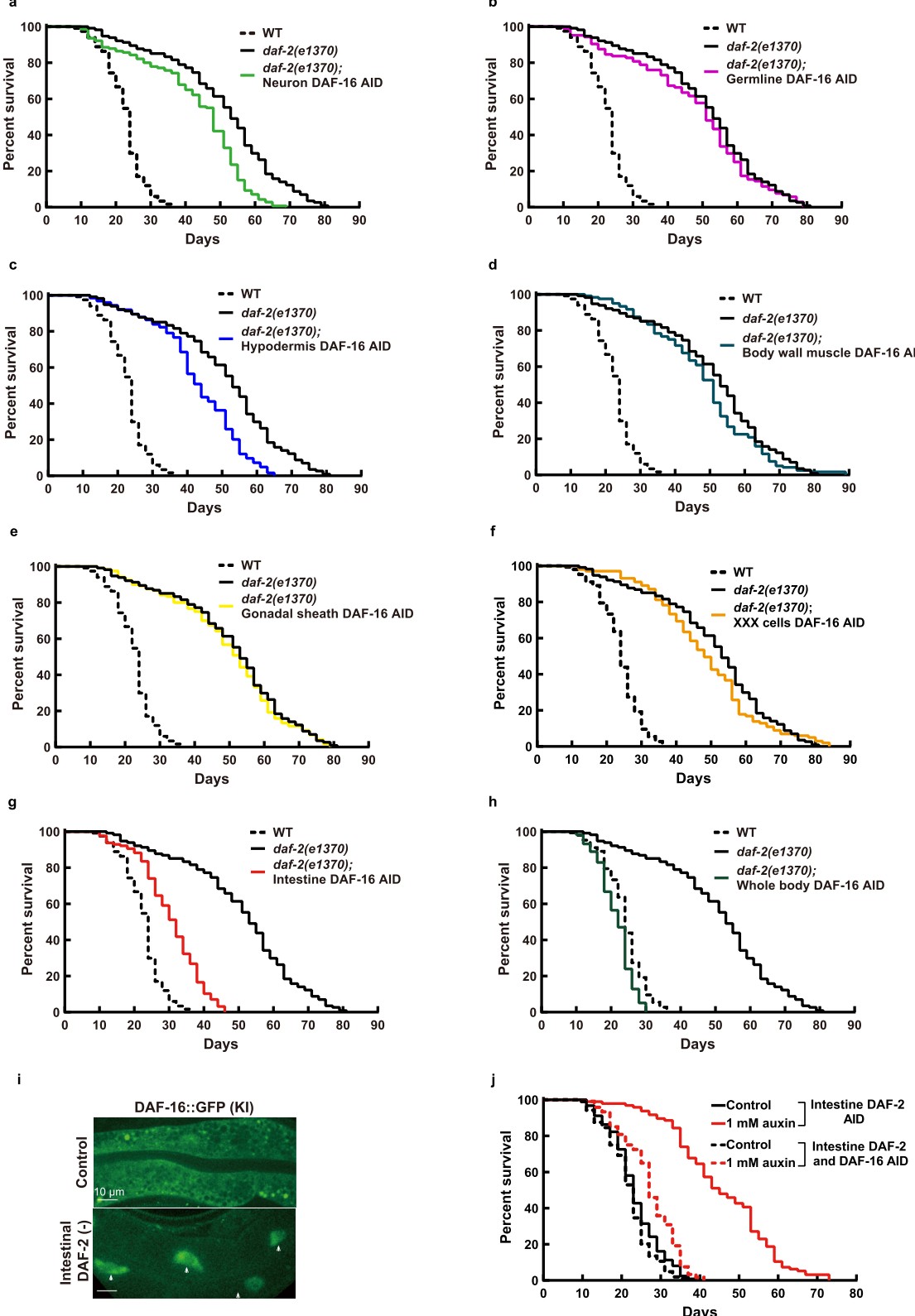

**Fig. 4 | Intestinal DAF-16 mediates *daf-2(e1370)* longevity. a** Degrading DAF-16 from the neurons shortened the *daf-2(e1370)* lifespan by 15.6% (*p* < 0.0001). **b** Degrading DAF-16 from the germline had no significant effect on the *daf-2(e1370)* lifespan (*p* = 0.347). **c** Degrading DAF-16 from the hypodermis shortened the *daf-2(e1370)* lifespan by 15.6% (*p* < 0.0001). Degrading DAF-16 in the body wall muscle (**d**), gonadal sheath (**e**), or XXX cells (**f**) had no effect on the lifespan of *daf-2(e1370)* (*p* > 0.05). **g** Degrading intestinal DAF-16 shortened the *daf-2(e1370)* lifespan by 40.1% (*p* < 0.0001). **h** Degrading DAF-16 in all tissues shortened the *daf-2(e1370)*

lifespan by 57.8% (*p* < 0.0001). **i** Endogenously expressed DAF-16::GFP accumulated in the intestinal nuclei upon degrading DAF-2 from the intestine. A similar pattern of expression was observed in three independent experiments. **j** Degrading intestinal DAF-16 largely suppressed the longevity induced by degrading DAF-2 from the intestine (red dashed line versus red solid line, *p* < 0.0001). *p* values are calculated by log-rank tests. See survival statistics in Supplementary Data 1. Source data are provided as a Source Data file.

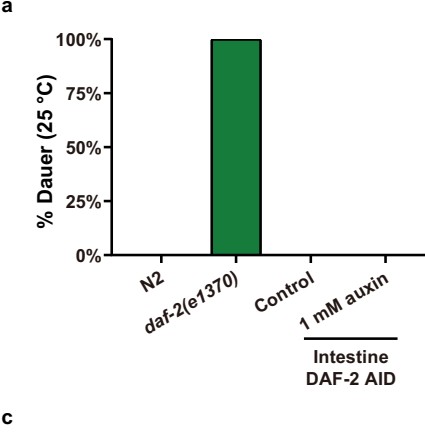

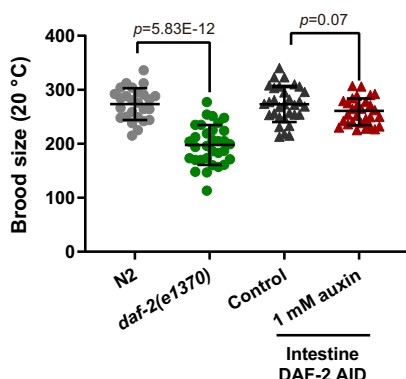

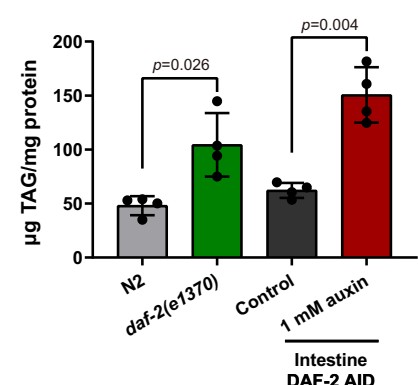

**Fig. 5 | Intestine-specific degradation of DAF-2 had no adverse effect on development and reproduction. a** Intestinal degradation of DAF-2 did not cause dauer arrest at 25 °C. **b** The intestinal DAF-2 AID worms developed faster than the *daf-2(e1370)* worms. Data are represented as mean ± SEM of three biological replicates. **c** The brood size of intestinal DAF-2 AID worms is comparable to that of the control animals. Data are represented as mean ± SD of two biological replicates.

*p* values are calculated by unpaired two-sample *t*-test. **d** Degradation of intestinal DAF-2 elevates the triacylglycerol (TAG) content by 2.4-fold relative to the control animals. Data are represented as mean ± SEM of four biological replicates. *p* values are calculated by unpaired two-sample *t*-test. Source data are provided as a Source Data file.

metabolism genes that are down-regulated in the *daf-2* mutant, extended WT lifespan by 19.0% or 23.6%, respectively (Fig. 6d). *M28.5/ snu-13* encodes a conserved protein functioning in rRNA processing and mRNA splicing[69], and *fib-1* encodes the *C. elegans* fibrillarin, which catalyzes 2′-O-methylation of pre-rRNA and U6 snRNA[70,71]. Lifespan extension by *fib-1* RNAi has been reported[64,72]. These data suggest that a reduction in RNA metabolism contributes positively to *daf-2* longevity.

In summary, intestine-specific degradation of DAF-2 helps identify core gene expression changes underlying the longevity phenotype of *daf-2* worms: down-regulation of RNA and protein metabolism. These are shared molecular signatures of longevity following IIS reduction by different means.

**Loss of intestinal DAF-2 induces systemic changes**

The finding that degrading DAF-2 specifically in the intestine nearly doubled *C. elegans* lifespan raised another question, namely, whether and how intestinal IIS affects other tissues. To address this question, we constructed tissue-specific GFP reporters in the intestinal DAF-2 AID strain, with which we isolated the cells of interest from day 1 adults either treated with auxin or not, for tissue-specific RNA-seq (Fig. 7a and Supplementary Fig. 13a, b). Principal component analysis (PCA) of the RNA-seq data showed clear distinctions between tissues and between treatments (Supplementary Fig. 13c).

From the isolated intestinal, hypodermal, neuronal, and BWM cells, we found respectively 508, 212, 209, and 26 DEGs following degradation of DAF-2 in the intestine (Supplementary Fig. 13d and Supplementary Data 4), with a moderate enrichment of Class I and

Class II DAF-16 targets among the hypodermal and neuronal DEGs (Supplementary Fig. 13e). These data demonstrate a clear cross-tissue effect of intestinal IIS on hypodermis and neurons. That only 26 DEGs were found in muscle cells echoes the finding that IIS in muscles is not important for IIS effects on lifespan.

Gene ontology (GO) analysis of DEGs of different tissues revealed an intriguing pattern (Fig. 7b). Among the ten up-regulated genes that responded to degradation of intestinal DAF-2, eight of the enriched GO terms seen in at least two tissue types are related to protein phosphorylation (Fig. 7b, left panel). GO terms enriched from the down-regulated genes are related predominantly to protein and RNA metabolism including translation, transcription, degradation of protein, and degradation of RNA (Fig. 7b, right panel), which re-confirms the finding from whole worm RNA-seq data (Fig. 6a, right panel, and Fig. 6b). The intestine displayed a higher degree of similarity with the hypodermis than with the neurons, but downregulation of structural constituents of ribosome and translation is a shared transcriptional response among all three tissues (Fig. 7b). Therefore, decreased protein synthesis, an important mechanism for IIS longevity[68], is propagated from the intestine to other tissues. Of note, Dietary Restriction (DR), which promotes longevity, also decreases protein synthesis[61,67,73].

Importantly, intestinal DAF-2 degradation induced nuclear accumulation of DAF-16::GFP in the hypodermis during the L1-L2 stage (Fig. 7c and Supplementary Fig. 13f), suggesting that hypodermal DAF-16 is activated. Class I DAF-16 targets, with an enrichment of the DAF-16-binding element in their promoter sequences[60], are more likely to contain direct DAF-16 targets than the Class II targets. So, from the hypodermal and neuronal DEGs (Supplementary Fig. 13d, e),

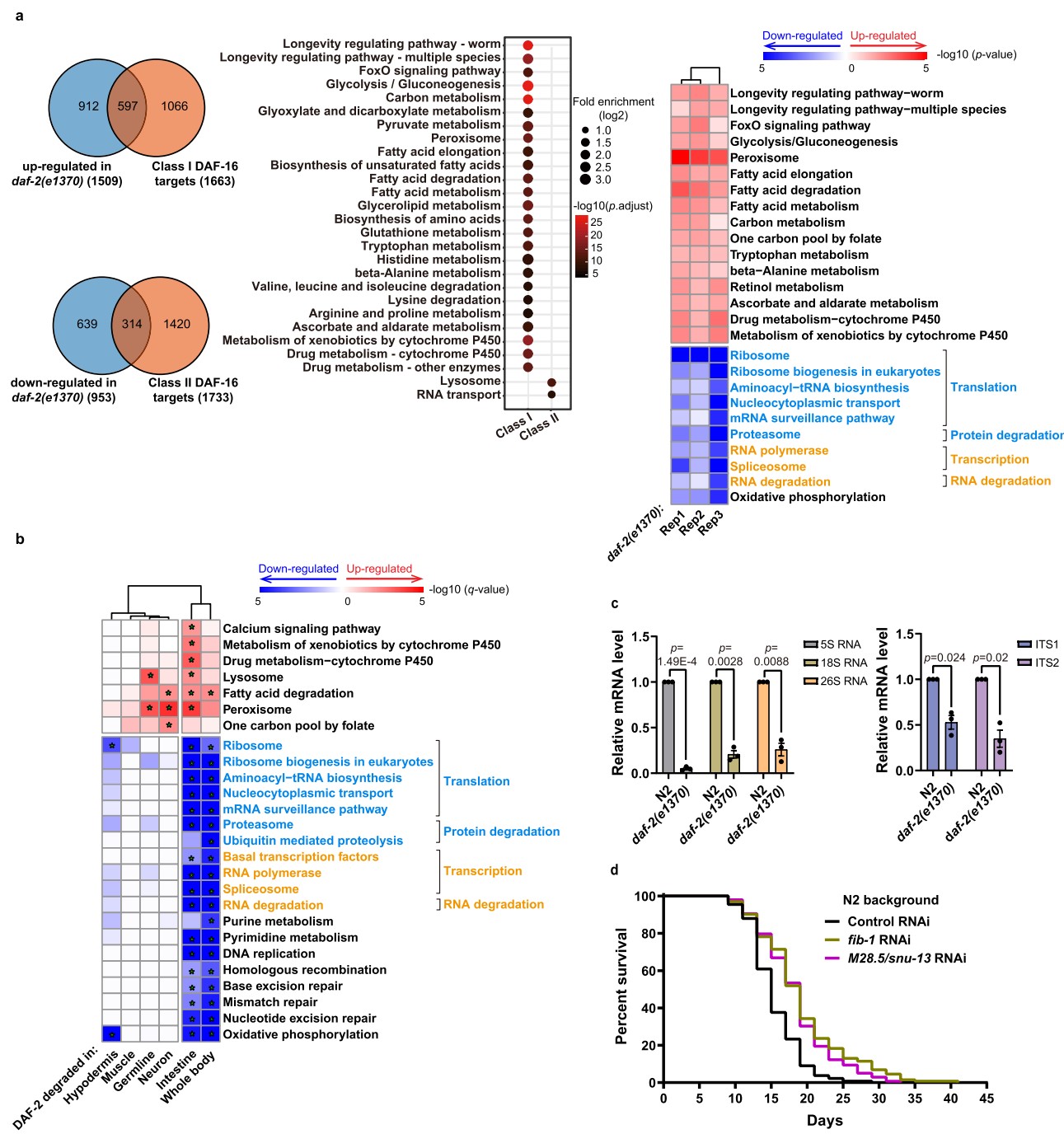

**Fig. 6 | Intestine-specific degradation of DAF-2 evokes a comparatively simple pro-longevity transcriptional program. a** Transcriptome analysis of *daf-2(e1370)* versus WT (N2) worms. Left panel, overlap of the canonical DAF-16 targets and DEGs (*q* value < 0.05) of *daf-2(e1370)*. Middle panel, enriched KEGG pathways among Class I and Class II DAF-16 targets by Over Representation Analysis (ORA). Only pathways with adjusted *p* value < 0.001 are shown. For multiple comparisons, adjustments were made with Benjamini–Hochberg (BH) method. Right panel, GSEA of *daf-2(e1370)* worms. Pathways with *p*-value < 0.01 are shown, and those related to protein metabolism (blue color) and RNA metabolism (orange color) are highlighted. See statistics in Source data and Supplementary Data 2. **b** GSEA analysis of worms subjected to tissue-specific degradation of DAF-2. Pathways with *q*-value < 0.01 (labeled by *) in at least one sample are shown, and those related to protein metabolism (blue color) and RNA metabolism (orange color) are highlighted. For multiple comparisons, adjustments were made with Benjamini–Hochberg (BH) method. **c** qRT-PCR analysis of ribosomal RNAs (left panel) as well as their precursors (right panel) in *daf-2(e1370)* and N2 worms. Data are represented as mean ± SEM of three biological replicates. *p* values are calculated by unpaired two-sample *t*-test. **d** Knocking down of genes related to RNA metabolism (*fib-1* and *M28.5*) by RNAi further extended the lifespan of WT worms. See survival statistics and biological replicates in Supplementary Data 1. Source data are provided as a Source Data file.

we selected four Class I targets and constructed promoter reporter lines. Indeed, they were induced outside the intestine following degradation of intestinal DAF-2 (Fig. 7c, Supplementary Fig. 13g, h). Among the four selected Class I targets, *dod-3* and *mtl-1* are likely direct targets of DAF-16[74–78]. We also verified the induction of *gst-30*, another DEG but not a Class I DAF-16 target, outside the intestine (Fig. 7c). In brief, these data indicate that DAF-16 in non-intestinal tissues is activated to some degree in the absence of intestinal DAF-2.

So far, we have shown that degradation of intestinal DAF-2 activates DAF-16 not only in the intestine (Fig. 4i), but also in other tissues

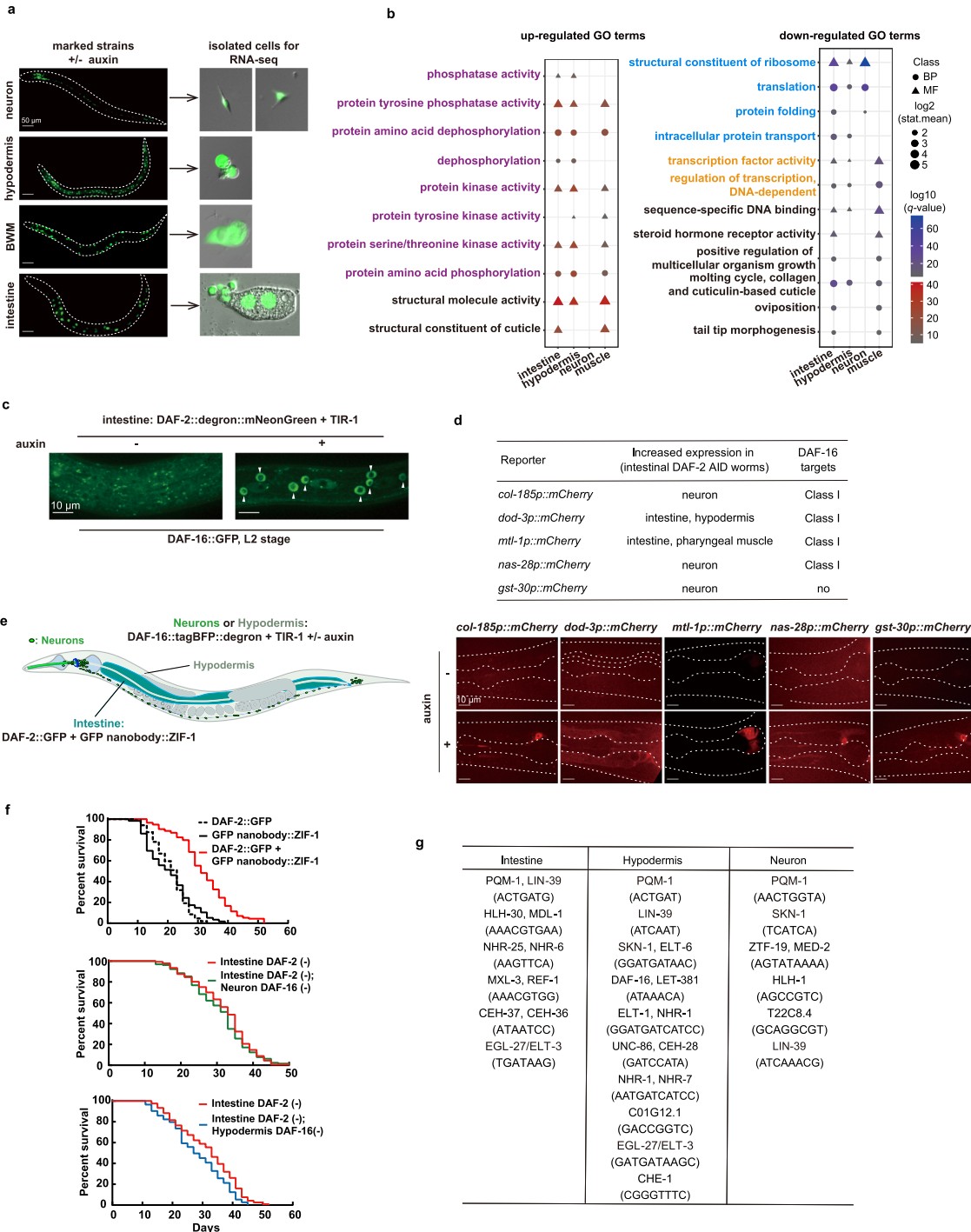

**Fig. 7 | Loss of intestinal DAF-2 triggered gene expression changes in other tissues through cross-tissue DAF-2 to DAF-16 signaling. a** Isolation of neurons, hypodermis, body wall muscle (BWM), and intestine cells from intestinal DAF-2 AID worms for RNA-seq. Left panel, tissue-specific transgenic reporters in the intestinal DAF-2 AID strain used to isolate tissue-specific cells. Neurons, labeled by *rgef-1p::NuGFP*; hypodermis, labeled by *dpy-7p::NLS^SV40^::GFP*; BWM, labeled by *myo-3p::NuGFP*; intestine, labeled by *ges-1p:: NuGFP*. Right panel, representative images of tissue-specific cells isolated by FACS for RNA-seq. A similar pattern of expression was observed in two independent experiments. **b** GSEA analysis of up-regulated and down-regulated GO terms enriched in each isolated tissue. GO terms with *q*-value < 0.01 in at least two tissues are shown. For multiple comparisons, adjustments were made with Benjamini–Hochberg (BH) method. **c** Degrading DAF-2 from the intestine induces DAF-16 nuclear accumulation in the hypodermis at the L2 larval stage. A similar pattern of expression was observed in three independent experiments. **d** Five DEGs selected to verify the cross-tissue effect of intestinal IIS on non-intestinal tissues by *mCherry* transgenic reporters in the intestinal DAF-2

AID strain. Representative images of each reporter are shown in the bottom panel. A similar pattern of expression was observed in three independent experiments. **e** Schematic of the combination of tissue-specific GFP nanobody-mediated ZIF-1 system and tissue-specific AID system to simultaneously degrade DAF-2 and DAF-16 in two different tissues, the intestine and non-intestinal tissues (neurons or hypodermis). **f** Lifespan phenotypes following simultaneous degradation of intestinal DAF-2 and non-intestinal DAF-16. Degrading intestinal DAF-2 by GFP nanobody-mediated ZIF-1 system extended lifespan by 51.8% (top panel, *p* < 0.0001). Degrading DAF-16 in the hypodermis (bottom panel, *p* = 0.003), but not in the neurons (middle panel, *p* = 0.337), moderately but significantly decreased the lifespan of the worms in which the intestinal DAF-2 level was reduced. *p*-values are calculated by log-rank tests. See survival statistics in Supplementary Data 1. **g** Motif enrichment analysis of transcriptional binding sites among 1-kb promoter sequence of the intestinal, hypodermal, and neuronal DEGs. See statistics in Supplementary Data 5. Source data are provided as a Source Data file.

(Fig. 7c, d and Supplementary Fig. 13e–h). This is consistent with previously proposed FOXO-to-FOXO signaling[39,74,79,80]. Using targeted protein degradation, we further investigated whether lifespan extension by intestinal DAF-2 degradation requires non-intestinal DAF-16. We knocked in a tagBFP::degron tag to the genomic locus of *daf-16*, thereby subjecting the expressed tagBFP::degron fusion protein to auxin-induced tissue-specific degradation (Fig. 7e). To degrade intestinal DAF-2 in the same animals, we knocked in a GFP sequence to the *daf-2* locus and expressed a GFP nanobody::ZIF-1 fusion protein under an intestine-specific promoter[81]. Unlike the AID system, GFP-mediated degradation of DAF-2 was incomplete, but enough to extend lifespan by 51.8% (Fig. 7f, top panel). We found that degrading DAF-16 in the hypodermis (Fig. 7f, bottom panel), but not in the neurons (Fig. 7f, middle panel), moderately but significantly shortened the lifespan of worms in which intestinal DAF-2 level was reduced. The DAF-16 binding site was also enriched among the hypodermal DEGs following the removal of intestinal DAF-2 (Fig. 7g). Therefore, we conclude that hypodermal DAF-16 is activated by the loss of intestinal DAF-2 and contributes to longevity.

From the tissue-specific RNA-seq data, we also found evidence suggesting the involvement of other TFs (Supplementary Data 5). The PQM-1 and SKN-1 binding sites were significantly enriched in 1-kb promoter sequence of the hypodermal and neuronal DEGs (Fig. 7g). These two TFs are required for *daf-2* longevity[60,82].

In summary, the data above demonstrate that the intestine is the primary action site for IIS to regulate lifespan. However, in addition to intracellular signaling from DAF-2 to DAF-16 within the intestine, DAF-16 in other tissues is also activated in response to reduced signaling from DAF-2 in the intestine and contributes to longevity.

## Discussion

In contrast to the diverse pleiotropic phenotypes of *daf-2* mutants, previous studies reported restricted and differing expression patterns of *daf-2*, either in neurons and XXX cells based on immunostaining[41] or in the germline based on in situ hybridization[83]. In this study, we found that in addition to neurons, XXX cells, germ cells, and vulval cells—which display the brightest DAF-2::mNeonGreen signal on the cell surface—the *daf-2* gene is also expressed in the intestine, hypodermis, muscles, and gonadal sheath. We believe that this expression pattern is accurate and complete for the following reasons. First, we engineered the genomic loci of the *daf-2* and *daf-16* genes to label their endogenous protein products with fluorescent tags (Figs.1 and 2), thereby avoiding artifacts associated with transgene reporters. Because all the DAF-2 isoforms share the same C-termini, and so do all the DAF-16 isoforms, the C-terminal mNeonGreen or GFP tag should label all DAF-2 or DAF-16 proteins. Second, we developed an ultra-sensitive tagging method (Fig. 1a), with which we found that most or all *C. elegans* cells express *daf-2* mRNA (Fig. 1c). This ubiquitous expression pattern of *daf-2* overlaps perfectly with that of its downstream TF *daf-16* and is consistent with the many pleiotropic phenotypes of *daf-2* mutants. For example, mutations of *daf-2* increase lipid storage in the intestine, promote dauer formation during development (which involves the hypodermis producing a dauer cuticle), and suppress induction of multiple vulvae[53]. Prior to this study, *daf-2* expression had not been detected in the intestine, hypodermis, and vulval cells, where the above phenotypes are found.

Anti-ageing by reducing neuronal IIS has been an attractive idea since expression of a *daf-2* transgene under a neuronal promoter, but not a muscle or intestinal promoter, was found to suppress the longevity phenotype of the *daf-2* mutant[40]. However, transgenic expression of *daf-16* under an intestinal promoter, but not a neuronal or muscle promoter, extended the lifespan of the *daf-2; daf-16* double mutant[39]. This is consistent with intestine-specific DAF-16 degradation shortening *daf-2* lifespan[84]. Recently, three studies using either tissue-specific knockout[38] or tissue-specific protein degradation[85,86] all report

that neuronal DAF-2 and intestinal DAF-2 have comparable effects on lifespan, as loss of either extends WT lifespan by 50% or so. Here, we showed that intestine or neuron-specific degradation of DAF-2 extended lifespan by 94.4% or 18.6%, respectively. Although we cannot completely exclude the possibility that leaky expression below the detection limit may result in an overestimation of the contribution of IIS in some tissues, a consensus is forming that intestinal DAF-2 has a substantial effect on lifespan regulation. However, there is no consensus yet regarding neuronal DAF-2.

To solve this controversy, each of the above experiments must be examined to ensure the absence or reduction of neuronal DAF-2 and the intactness of intestinal DAF-2. None of the above studies showed direct evidence of the latter, and only this study showed direct evidence of the former. This study is the only one so far that determined the expression pattern of *daf-2*, and the DAF-2::mNeonGreen or *daf-2::degron::mNeonGreen* KI strain should be a useful tool to visualize the effect of *daf-2* knockdown, knockout, or DAF-2 protein degradation in neurons. Pertaining to the intactness of intestinal DAF-2 in our degradation experiments of neuronal DAF-2, although the weak intestinal DAF-2::mNeonGreen signal did not provide a direct readout, we reason that intestinal DAF-2 was not affected when neuronal DAF-2 was degraded based on the following: (1) the *rgef-1* promoter showed good neuron specificity (Supplementary Fig. 4c); (2) neuronal DAF-2 was verifiably degraded (Supplementary Fig. 5a and 8); (3) this treatment extended lifespan by 18.6%; (4) intestine-specific degradation of DAF-2 extended lifespan by 94.4%; and (5) if the 18.6% lifespan extension had resulted from an unintended loss of intestinal DAF-2, it would further refute the idea of IIS regulating lifespan from neurons. Lastly, suspecting that the conflicting results might be related to the different neuronal promoters used, we compared side by side the *rgef-1* promoter used in this study and the *rab-3* promoter used in two other studies[85,86]. As shown in Supplementary Fig. 14, *rab-3p::NuGFP* showed leaky expression in the intestine[87], whereas *rgef-1p::NuGFP* did not. This suggests that the seemingly strong longevity phenotype of neuronal DAF-2 AID[85,86] is induced partly and unintentionally by intestinal DAF-2 AID, due to leaky expression of TIR-1 under the *rab-3* promoter.

For the above reasons, we believe that this study clarifies the confusion regarding which tissue initiates the anti-aging signal in the long-lived IIS mutants. The study by Libina et al correctly points out that it is the intestine[39], but the evidence displayed is not strong enough, nor complete, in the following aspects: 1) intestine-specific expression of DAF-16 from extrachromosomal arrays in the *daf-2; daf-16* double mutant worms restored longevity about halfway, leaving it possible that there may be another tissue which is also a major site of action of IIS in lifespan regulation; (2) no tissue-specific functional analysis of *daf-2* to either confirm or reject the conclusion of an earlier study[40] that the *daf-2* gene acts in neurons to regulate lifespan; (3) did not address directly what might be the cause of the conflicting results from different studies. We are glad that the Libina study is backed up by this one and we are hopeful that research can move on from this sticky point. Reflecting on the intestinal versus neuronal IIS effects on lifespan, it is worth noting that *daf-2* RNAi induces consistently a longevity phenotype[32]. It is well known that *C. elegans* neurons are refractory to RNAi[88,89], whereas other cells including the intestinal cells are susceptible. Given what we find in this study, longevity by *daf-2* RNAi can be well explained: the intestine responds readily to *daf-2* RNAi and the intestine happens to be the primary site of action for IIS in lifespan regulation.

Related to this, mechanistic studies of lifespan extension by certain neuronal changes including the mitochondrial unfolded protein response (UPR^mt), expression of XBP-1s, and activating neuronal CRTC-1, often identify the intestine as the responsive organ[90–94]. Many signaling pathways that influence lifespan converge on DAF-16, including IIS, AMPK signaling, mitochondrial signaling, CRTC-1, and germline signaling[1,90]. Activation of intestinal DAF-16, in particular, underlies a

growing number of lifespan-extending conditions, including germline ablation[39,47,95], overexpression of HSF-1 in neurons[96], temperature-dependent lifespan regulation by IL1 and ASJ neurons[97], and the classic case of IIS reduction[38,39].

Lifespan extension by activating intestinal DAF-16 is not unique to *C. elegans*. In mammals, IIS bifurcates into insulin signaling and IGF-1 signaling. Both IR and IGF-1R are widely expressed in mammals (https://www.proteinatlas.org and http://www.informatics.jax.org). Heterozygous brain-specific knockout of IGF-1R or insulin receptor substrate-2, which transmits signaling from both IR and IGF-1R, has been reported to extend lifespan in mice along with other phenotypes such as growth retardation or insulin resistance[34,36]. The worm intestine, in addition to being a digestive tract, doubles as an adipose tissue. In mice, adipose tissue-specific knockout of IR (FIRKO mice) extends lifespan[11]. Unlike intestinal DAF-2 AID, which more than doubles fat storage in *C. elegans* (Fig. 5d), FIRKO reduces the fat mass of mice by half or more[11]. Intestinal epithelium-specific knockout of IR alleviates insulin resistance in aged animals, but the effect on lifespan is not reported[98]. In both knockout mice, the FoxO TFs—the mammalian counterpart of DAF-16—are activated in the tissues lacking IR. In the future, it would be interesting to find out whether intestinal epithelium-specific degradation or deletion of IR extends lifespan of mice.

Studies of a complex question such as aging in a simple invertebrate model serve to provide directions to studies of the same question in complex vertebrate animals. In this regard, the *C. elegans* research on lifespan regulation by IIS has been a showcase example[1]. However, earlier findings from *C. elegans* on how to extend lifespan without adverse effects on development and reproduction fall short of providing specific directions, as explained below. Potentially, one way to getting only the good effect is to look for special mutations on IIS genes, but this has not met with success. Two null alleles of *age-1* can extend *C. elegans* lifespan 9–10 times, which is the most dramatic longevity phenotype of any IIS mutant recorded to date, but those worms are completely sterile, severely retarded in development, and strongly prone to forming dauers[99]. The canonical *daf-2(e1370)* allele, which doubles *C. elegans* lifespan, is so far the best one, but it is not free of pleiotropic phenotypes in development and reproduction (Fig. 5 and refs. 33, 100). Another way is through *daf-2* RNAi, but to what extent IIS activity is reduced by *daf-2* RNAi is not known and the extent of lifespan extension by *daf-2* RNAi is variable from experiment to experiment, but never approaching that by *daf-2(e1370)*. With either mutations or RNAi, the problem encountered is similar: it is unknown whether extraordinary longevity at no fitness cost is achievable by reducing IIS globally, and if so, by how much and how to modulate IIS to that perfect level. Based on the findings in mice and the findings in *C. elegans*, as described above, we suggest that intestine-specific removal or reduction of IIS is a potentially promising approach to achieve mammalian longevity with little to no adverse effects on development or reproduction.

Like the *daf-2(e1370)* and the whole-body DAF-2 AID worms, the longevity of intestinal DAF-2 AID worms is partly attributable to elevated protection against bacterial infection (Supplementary Fig. 9 and refs. 55–57). Enhanced resistance to bacterial infection of the intestine is most likely a local effect of DAF-2 AID in the intestine, that of the pharynx is not. The part of the lifespan extension that cannot be explained by enhanced innate immunity is more likely than not a systemic effect of a longevity signal sent from the DAF-2-deficient intestine to the entire body.

Interestingly, the life-shortening effects of gentamicin in *daf-2(e1370)* worms[55] was once again observed in DAF-2 AID worms, suggesting that gentamycin has antagonistic effects on survival: promoting survival by preventing infection, but also shortening lifespan when worms are fully resistant. One possible explanation is that gentamicin itself is mildly toxic to *C. elegans*, perhaps more so in the *daf-2(e1370)*

mutant and DAF-2 AID worms. Another possibility is that gentamycin makes the bacteria nutritionally inadequate, as worms grow poorly on autoclaved bacteria[101]. One might be able to avoid these issues by using other means to prevent bacterial proliferation (e.g., other antibiotics[102,103], UV irradiation[104], or paraformaldehyde[105]).

It is noticed that IIS and DR share commonalities: both are evolutionarily conserved mechanisms that regulate ageing, and both involve nutrient-sensing mechanisms connected to regulation of protein synthesis and autophagy[61,62,68,106,107]. It is generally agreed upon that DR reduces mTOR signaling[108–110]. However, whether DR or inhibition of mTOR intersects with IIS has remained unclear, since most forms of DR extend lifespan in a DAF-16-independent manner[49,111,112]. Conversely, there is no direct evidence of reduction of mTOR activity in the long-lived IIS mutants[113]. So, lifespan regulation by IIS and DR seem to invoke distinct signal transduction pathways[112,114], but certain downstream biological processes such as translation are targeted by both. Our findings in this study call attention to what is likely the common upstream change—altered nutrient supply to various tissues. As discussed above, the intestine is frequently found to be the responsive tissue/organ downstream of many lifespan-extending conditions which are not limited to IIS and DR. We hope that the findings from this study will be helpful to focus the ever-expanding directions of anti-ageing research.

## Methods

### *Caenorhabditis elegans* maintenance
Nematodes were maintained at 20 °C on standard nematode growth medium (NGM) agar plates seeded with *Escherichia coli* OP50 unless otherwise stated[115].

### Strain construction
1) Transgenic strains
To generate transgenic animals carrying extrachromosomal arrays (hqEx), 5–50 ng/µl of the indicated plasmid was injected to the gonad using standard method.
2) Knock-in strains
CRISPR engineering for all knock-ins was performed by microinjection using the homologous recombination approach[116]. The injection mix contained at least two Cas9-sgRNAs expressing plasmids (50 ng/µl for each), a selection marker pRF4 (*rol-6(su1006)*, with roller phenotype) (50 ng/µl), and a homologous recombination plasmid (50 ng/µl). To generate the sgRNA plasmids, primers were designed with the CRISPR DESIGN tool (https://zlab.bio/guide-design-resources) and inserted into the pDD162 vector (Addgene #47549) using the site-directed mutagenesis kit (Toyobo). To generate the homologous recombination (HR) plasmids, two homologous arms (-1000 bp each) corresponding to the 5′- and 3′-sides of the insertion site, respectively, were cloned into the vector. All injection plasmids were purified with PureLink PCR Micro kit (Invitrogen) and injected into the gonad of young adult hermaphrodite worms using standard method. F1s with roller phenotype were singled on a new NGM plate and allowed to produce sufficient offspring. Successful knock-in events were screened by PCR genotyping from independent F1 transgenic animals' progeny that did not display roller phenotype, and further confirmed by DNA sequencing.

Strains and sgRNAs used in this study are listed in Supplementary Data 6 and 7, respectively.

### Confocal imaging
Confocal image of Fig. 1c (left panel) was captured by ZEISS LSM 880 microscope equipped with a 63×, 1.4 numerical aperture oil-immersion objective as Z-stacks of 1 µm-thick slices under the lambda-mode (settings: pixel dwell: 2.06 µs; average: line 1; master gain: 750; pinhole size: 91 µm; filter: 411–695 nm; beam splitter: MBS 488; lasers: 488 nm, 30%). Images were processed using ZEN 2 software (Carl Zeiss Inc.).

All other confocal images were taken using the spinning-disk microscope (UltraVIEW VOX; PerkinElmer) equipped with a 60×, 1.4 numerical aperture oil-immersion objective, except for those in Fig. 7a (left panel) and Supplementary Fig. 13g, h taken using 10× objective. Images were viewed and processed using Volocity 6.4 software (PerkinElmer).

Worms were anaesthetized using 1 mM levamisole hydrochloride in water on 3% agarose pads on glass slides. In all the imaging studies, images within the same figure panel were taken with the same parameter and adjusted with identical parameters using ImageJ software.

### Auxin treatment
Auxin treatment was performed by transferring worms to OP50-seeded NGM plates containing 1 mM auxin as previously described[54]. Briefly, the 400 mM natural auxin indole-3-acetic acid (Alfa Aesar) stock solution in ethanol was prepared freshly. Then, the stock solution was added into the NGM agar cooled to about 50 °C in water bath before pouring plates. For all auxin treatment experiments, 0.25% ethanol was used as a control. 100 μl of OP50 overnight culture was seeded onto the auxin or control NGM agar plates (60 × 15 mm). Aluminum foil was used to protect the auxin-containing plates from light, and then kept plates at room temperature for 1–2 days before use.

### Lifespan analysis
All lifespan assays were performed at 20 °C unless otherwise stated. Strains were synchronized by allowing 40 gravid adults to lay eggs for 4 h on OP50-seeded NGM plates at 20 °C. Then, ~150 worms at early adulthood stage were transferred on ten fresh OP50-seeded NGM plates containing 1 mM auxin or 0.25% ethanol. Animals were transferred to new plates every day until the end of reproductive period, after which worms were transferred to fresh plates every week. For lifespan assays involved degrading DAF-2 throughout the whole body (Fig. 3h and Supplementary Fig. 9), 50 μg/ml 5-fluoro-deoxyuridine (FUDR) was also added into the NGM plates to prevent its internal egg hatching phenotype from interfering with the lifespan measurement. In these experiments, worms were transferred to fresh plates every 4 days until death.

For lifespan assays of feeding DAF-2 AID worms with dead bacteria, OP50 *E. coli* were killed by gentamicin treatment as previously described[55]. Briefly, OP50 cultures were allowed to grow to OD of 1.0 following by 120 μg/ml gentamicin treatment at 37 °C for 5 h. The OP50 solution was then washed, concentrated 50×, and seeded on NGM plates containing 10 μg/ml gentamicin the day before adding worms. Killing was confirmed by streaking lawns on antibiotic-free LB plates. Bleached eggs were placed and grown on these dead OP50-seeded NGM plates. Then, worms reaching late L4 stage were transferred to NGM plates containing 10 μg/ml gentamicin, 50 μg/ml FUDR, and 1 mM auxin/0.25% ethanol seeded with dead bacteria for the remaining life. In this experiment, worms were transferred to fresh plates every 4 days until death.

Live worms were scored every 2 days. A worm was considered as death if it showed no response to the gentle touch with a platinum wire on head and tail. Worms that had internally hatched larvae ("bagged") or ruptured vulvae ("exploded") or crawled off the agar surface or became contaminated were censored from the analysis. Statistical analyses were performed using IBM SPSS Statistics 20 software. *p* values were calculated using the log-rank (Mantel-Cox) method.

### DAF-16 nuclear translocation
To analyze DAF-16 nuclear localization, worms were cultured at 20 °C on NGM plates containing 0.25% ethanol or 1 mM auxin from eggs laid within a 4-h period and imaged at the indicated stage in the figure (Fig. 4i, day 1 of adulthood; Fig. 7c, L2 larval stage; Supplementary Fig. 13f, L1 larval stage). As soon as worms were removed from incubation, they were mounted on slides and imaged immediately.

### Dauer assay
For the dauer assay of N2 and *daf-2(e1370)* worms, 25-30 gravid adults were allowed to lay eggs for 1 h on OP50-seeded standard NGM plates at 20 °C. For the dauer assay of intestinal DAF-2 AID worms, eggs were laid on NGM plates containing 0.25% ethanol or 1 mM auxin. After picking off the adults, the resulting synchronous population was transferred to 25 °C or 27 °C and scored dauer formation after 72 h post hatch.

### Developmental assay
We semi-synchronized the worms by allowing 30 gravid adults to lay eggs for 2 h at 20 °C and then removing out the adult worms. After 64 h, the developmental state of the worms was assessed.

### Brood size assay
16 synchronized L4 stage worms were singled on individual NGM plates seeded with OP50. The animals were transferred to fresh plates every 24 h for 4–5 days. Worms that crawled off the plates, bagged or exploded were censored from the analysis. The number of hatched worms was counted 2 days later. The *p* values were determined by Student's *t* test.

### TAG measurement
TAG measurement was performed as previously described[117]. Briefly, 8000 synchronized L1 larval worms were placed on each 10 cm OP50-seeded NGM plate with or without auxin and collected at late L4 stage. One-eighth of worms was taken out to extract total soluble protein, and the protein concentration was quantified using the BCA Protein Assay Kit (Pierce). The sample for TAG measurement was homogenized and 20 μg tri-C17:0 TAG (Nu-Chek) was added as an internal calibration standard, followed by the extraction of total lipid. TAG was separated from total lipid on a thin-layer chromatography plate, then transmethylated with 2 ml methanol and 50 μl sulfuric acid to prepare fatty acid methyl esters (FAMEs). FAMEs were re-dissolved in 2 ml pentane and chromatographed using a GC/MS instrument (SHIMADZU, QP2010 Ultra) with a DB-23 GC column (Agilent, 122-2332). FAME peaks were identified according to the fatty acid standards and integrated to calculate the TAG amount, which was finally normalized by the protein amount.

### qRT-PCR
Total RNAs were extracted from N2 and *daf-2(e1370)* worms on adult day 1 using TRIzol (Invitrogen), followed by removing contaminant DNA by DNase I treatment. The cDNA was synthesized by using a reverse transcription kit (Takara Bio). qPCR was carried out on an ABI 7500 Fast real-time PCR system using a Takara real-time PCR kit (SYBR Premix Ex TaqTM II). mRNA levels of *pmp-3* and *act-1* were used as the internal control.

### Preparation worm samples for RNA sequencing at the whole worm level
Six AID worm samples of degrading DAF-2 specifically in the intestine, neuron, hypodermis, muscle, germline, or whole body and one control worm (MQD2428(*hq363[daf-2::degron::mNeonGreen]*)) were prepared for RNA sequencing with three biological replicates. Synchronized L1 worms were initially cultured on the standard high-growth (HG) plates supplemented with OP50 for 24 h at 20 °C to prevent entering the dauer stage, and then transferred onto HG plates containing 1 mM auxin to degrade the DAF-2 protein in different tissues. Worms were washed with M9 buffer and harvested on adult day 1.

### Isolation of tissue-specific cells from the intestine-specific DAF-2 AID worms by FACS
Synchronized day 1 adult transgenic worms with GFP-labeled neurons, body wall muscle, hypodermis, or intestine (*rgef-1p::NuGFP,*

*myo-3p::NuGFP*, *dpy-7p::NLS::GFP*, and *ges-1p::NuGFP*) were prepared for cell isolation. Cells were isolated following the procedure described previously[118,119] with minor modifications. Briefly, worms were subjected to SDS-DTT treatment, proteolysis, and mechanical disruption. Cell suspensions were gently passed over a 5 μm syringe filter (Millipore) for neuron cell isolation; 20 μm filter (PluriSelect) for muscle and hypodermal cell isolation. To isolate intestinal cells, cell suspensions were passed through a 40 μm cell strainer (Falcon) and then spun at 800 × g for 3 min in a tabletop centrifuge. The filtered cells were diluted in PBS/20% FBS and sorted using BD FACS Aria II (BD Biosciences). Gates for detection were set by comparison to MQD2428 (*hq363*[*daf-2::degron::mNeonGreen*]) cell suspensions prepared on the same day from a population of worms synchronized alongside the experimental samples. Positive fluorescent events were sorted directly into tubes containing TRIzol LS (Invitrogen) for subsequent RNA extraction.

### RNA sequencing and data pre-processing
RNAs were isolated using a standard TRIzol/chloroform/isopropanol method, RNA quality and quantity were assessed using the Agilent 2100 Bioanalyzer RNA Pico chip (Thermo Fisher Scientific). For whole worm RNA-seq, libraries were prepared using the NEBNext Ultra RNA library Prep Kit (NEB) according to the manufacturer's instructions, and then sequenced using an Illumina HiSeq X Ten System in the paired-end mode (2 × 150 bp) through the service provided by Bionova. To construct RNA-seq libraries of FACS isolated tissue-specific cell samples, RNAs (200–500 pg) were amplified with oligo-dT, then reverse transcribed to cDNA based on polyA tail. The template was switched to the 5′ end of the RNA and the full-length cDNA was generated by PCR. Purified cDNA was fragmented into small pieces with fragment buffer by PCR, and the product was purified and selected by the Agencourt AMPure XP-Medium kit (Thermo Fisher Scientific). cDNA was quantified by Agilent Technologies 2100 bioanalyzer. The double stranded PCR product undergo QC step was heat denatured and circularized by the splint oligo sequence. The single strand circle DNA (ssCir DNA) was formatted as the final library. The final library was amplified with phi29 (Thermo Fisher Scientific) to make DNA nanoball (DNB) which had more than 300 copies of one molecular, DNBs were loaded into the patterned nanoarray and single end 100 bases reads were generated on BGISEQ500 platform (BGI-Shenzhen, China).

FASTQC (version 0.10.1) was used to inspect the quality of the raw sequencing data. The raw data were filtered to remove primers, contamination, and low-quality reads, then the Illumina adapter sequences were trimmed to obtain the clean sequencing data. The reads were aligned to the *C. elegans* reference genome (wbcel235.97) using HISAT2 (version 2.1.0)[120] with Ensembl gene annotations (using default parameters). Mapped reads that overlap with coding gene features were counted using featureCounts (version 1.6.5)[121]. The total number of mapped clean reads for each library ranged from 25 to 35 million.

### Gene differential expression (DE) analysis
RNA-seq raw counts were analyzed in RStudio (version 1.1.456) with version 3.6.1 of R. For the whole worm RNA-seq data, The R package DESeq2 (version 1.24.0)[122] was used for data normalization and differential expression analysis. Data quality was assessed by hieratical clustering of samples and principal component analysis (PCA). Differentially expressed genes between auxin treatment and control samples were determined using the *deseq* function which is based on the Negative Binomial (Gamma-Poisson) distribution. For the tissue-specific RNA-seq data, the R package edgeR (version 3.26.8)[123] was used for multidimensional scaling, PCA, and differential expression analysis. Four outliers (replicate 5 of neuron cell control sample, replicate 1 of intestine cell auxin treatment sample, replicate 1 of hypodermis cell

control sample, and replicate 2 of muscle cell auxin treatment sample) were removed from further analysis. Differentially expressed genes between auxin treatment and control samples were determined using the *glmQLFit* function (using the parameter "robust=TRUE") which is based on the quasi-likelihood F-test.

### Gene Ontology (GO) and KEGG pathway enrichment analysis
The R package gage was used to identify enrichment of GO terms and KEGG pathways which are based on a Generally Applicable Gene-set Enrichment (GAGE) method (version 2.34.0)[124]. The package cluster-Profiler (version 3.12.0)[125] was used to analyze the over-represented KEGG pathways within up- and down-regulated genes. GO term or KEGG pathway with an adjusted *p*-value or *q* value < 0.01 was defined as significantly changed unless otherwise noted. Data visualization was performed using the R package ggplot2 (version 3.3.2; Wickham H (2016)).

### TF binding motif analysis
1.0 kb promoter regions upstream of the DE genes from tissue-specific RNA-seq data were retrieved online from WormBase using the Parasite Biomart tool. The motif matrices were identified using RSAtools[126], and then analyzed using footprintDB[127] to identify the potential transcription factors predicted to bind to similar DNA motifs.

### Quantification and statistical analysis
Statistical analyses of lifespan assays were performed via the IBM SPSS Statistics 20 software. Detailed description of tests performed to determine statistical significance is included in figure legends. *, $p < 0.05$; **, $p < 0.01$; ***, $p < 0.001$; ****, $p < 0.0001$; ns, $p > 0.05$.

### Reporting summary
Further information on research design is available in the Nature Research Reporting Summary linked to this article.

## Data availability
All the RNA-seq raw data generated in this study have been deposited to NCBI under BioProject ID PRJNA770129. The *C. elegans* databases used in this study were WormBase release (wbcel235.97). Lifespan data are collected in Supplementary Data 1. Transcriptomics data are summarized in Supplementary Data 2, 3, and 4. Motif analysis data are included in Supplementary Data 5. Strains used in in this study are shown in Supplementary Data 6. sgRNA sequences used for constructing knock-in strains and primers used for quantitative RT-PCR are listed in Supplementary Data 7. Genome editing worm strains used in this study have been deposited at CGC, and all other relevant strains, plasmids, or data are available from the corresponding author on request. Source data are provided with this paper.

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

## Acknowledgements

We thank the Caenorhabditis Genetics Center (CGC), which is supported by the NIH Office of Infrastructure Programs (P40 OD010440), for providing worm strains. We also thank Drs. Guang-Shuo Ou and Xiao Liu for providing us worm strains and plasmids. This work was supported by Ministry of Science and Technology of China (2014CB84980001 to M.-Q.D.), Beijing Municipal Science and Technology Commission (a fund for cultivation and development of innovation base), and National Natural Science Foundation of China (NSFC-ISF 32061143020 to M.-Q.D. and NSFC 32171169 to S.O.Z.).

## Author contributions

Y.-P.Z., W.-H.Z., and M.-Q.D. designed the experiments and interpreted the data. Y.-P.Z. and W.-H.Z. performed most of the experiments and data analysis. P.Z. performed the qRT-PCR experiments and lifespan assays related to RNA metabolism. S.O.Z supervised Q.L. to conduct the TAG quantification experiment. Y.S. performed the 3D imaging experiment. J.-W.W. provided some supports on bioinformatics analysis. T.C. and C.Z. were involved in the interpretation of some of the data. Y.-P.Z., W.-H.Z., and M.-Q.D. drafted and revised the manuscript. M.-Q.D. supervised this study.

## Competing interests

The authors declare no competing interests.
