## [Peer Review File · Nature Communications]

Removal of intestinal DAF-2 nearly doubles lifespan but hardly affects development and reproduction of *Caenorhabditis elegans*REVIEWER COMMENTS

Reviewer #1 (Remarks to the Author):

In this study, Zhang et al. exploit the versatility of Crispr editing and the AID technology to bring clarity to an old discussion: which are the tissues relevant to the strong longevity phenotype of the IIS mutant *daf-2(e1370)*? First they show that both DAF-2 and its target DAF-16 are ubiquitously expressed throughout the worm body. Tissue-specific AID degradation of both proteins consistently show that they are required in the intestine to modulate longevity - neurons, hypodermis and germline seem to play a minor role. Intestinal degradation of DAF-2 results in a much narrower range of phenotypes (including longevity) compared to the pleiotropic effects of *daf-2(e1370)* mutation. Transcriptomics confirms that reduced protein and RNA metabolism may be central to worm longevity. Finally, IIS induced changes in the intestine also triggers a transcriptional response in other tissues.

This manuscript is very well written and presents a solid data set that includes the necessary checks and controls. Although some parts may seem a mere confirmation of some existing studies (which are appropriately cited by the authors), this study is relevant as it finally untangles some conflicting or hard-to-interpret data from the past. It is a big leap forward in the understanding of the role that tissues play in IIS-induced longevity in *C. elegans*. Therefore, I recommend acceptance of this manuscript after minor revision based on the specific comments below.

Specific comments

Line 74: The authors make a point that studying the expression pattern of poorly expressed genes or genes with complicated structures is difficult. However, they should mention the existence of alternatives such as cell- or tissue-specific transcriptomics (datasets are publicly available in on-line databases, e.g. Gexplore, CenGen). This work already showed broad expression of *daf-2* and *daf-16* in virtually all tissues.

Line 197: At several instances in the manuscript, the authors suggest similar downstream mechanisms of lifespan extension for IIS and dietary restriction – which seem to make sense based on the presented data. However, at first glance, the mechanistic link between IIS and DR may look contradictory to the earlier findings showing that DR-induced lifespan extension is often independent of DAF-16 (nicely summarized in Greer and Brunet, 2009 – Aging Cell). The authors may want to clarify this apparent contradiction in the discussion section (e.g. by stating that DR regimens may bypass intestinal IIS and directly cause nutritional scarcity in the worm body).

Line 210: Did the authors make a calculation mistake? According to the data in table S1 (and visual confirmation in figure 4G) the intestinal DAF-16-mediated lifespan extension of *daf-2(e1370)* is 'only' 72.2%, not 90.3%. Since this is one of the major findings of the paper, this number is relevant. I calculated this percentage as $100 * (1 - (30.50 - 22.63) / (50.95 - 22.63))$.

Line 244: This remarkable phenotype needs some discussion. The authors hypothesize that DR and IIS may have similar downstream molecular mechanisms to support longevity (assuming that a DAF-2 deficient intestine may provide the body with less nutrients?). How can this not have any repercussions on fertility? It is well-known that all kinds of DR negatively affect fertility. How can this be reconciled with this fully fertile, yet long-lived, intestinal DAF-2-deficient worm?

Line 291-302: This paragraph is not very clear. It is based on survival experiments with a limited number of worms (a single lifespan experiment with less than 100 worms). It looks like all knockdowns have a similar but modest lifespan extending effect which seems to support the claim that interfering with rRNA maturation extends lifespan. However, likely due to limited sample size, this effect is borderline non-significant in one case, and significant in the other two cases. This, despite the similar size effect (13-17%), makes this tiny data set hard to interpret. Moreover, it is not clear to me what the point is to test this in a *daf-2* background. Transcriptomics data suggest ribosome biogenesis is down in *daf-2* and this may (partially) cause longevity. Interfering with rRNA maturation

further increases daf-2 lifespan (as it likely does in wild type to a similar extent). Can you conclude anything specific from that? The hypothetical case where only lifespan extension would be observed in WT but not (further) in daf-2 would be more suggestive that daf-2 longevity benefits from reduced rRNA maturation/ribosome synthesis. I am not convinced that this little data set adds to the story.

Line 411: It is important to note that the entire study is dependent on the absolute tissue-specificity of the TIR1 promoters used. Some leaky expression may have large effects on the resulting phenotypes and final interpretation (e.g. rgef-1 is also expressed in the coelomocytes and the pharyngeal gland, lim-7 is expressed in many cell types, including neurons). Nevertheless, I appreciate the extensive controls in figures S4 and S5, but care should be taken and this should be mentioned.

Line 435: in this paragraph, the authors discuss the relevance of their findings and refer to IIS in a context beyond *C. elegans*. What I missed here is a critical note about the functional diversification of the IIS in mammals, where it has evolved into separate IGF and Ins signalling. A reflection on this (and where to act for possible future anti-ageing treatments) would have been more complete.

Typos and small errors

Line 394, 432: this study has been published now in eLife.

Figure 3: For tissue comparison, it would be better to use the same time scale for all panels (0-80 days)

Figure 5D: Legend Y-axis: ug TAG/mg protein ('1' can be deleted)

Figure 6 & S7: Avoid using red for the text (ribosome related pathways), this is confusing as 'red' also means 'upregulated' in this figure.

Figure S4B: mRuby (not mRubby, twice) – Pdpy-7 (not Pdyp-7)

Figure S8B: 'Events', why is this a range? Is this number of sorted cells over different replicates?

Reviewer #2 (Remarks to the Author):

The mechanisms by which insulin/IGF-1 signaling control aging in *C. elegans* is a topic that biogerontologists have been laboring over since the late 1980s, and the discovery of the long-lived age-1 mutant, such that a great deal is already known about the subject. This study by Zhang et al. uses new experimental tools, particularly CRISPR-Cas9, to address anew a number of old questions about this topic in a way that brings it into better focus. New methods are used to generate better reporter constructs for studying daf-2 and daf-16 expression, yielding a better understanding of daf-2 expression. Tissue specific degron mediated degradation of daf-2 and daf-16 confirm the importance of the intestine as a major site of action in IIS effects on lifespan, and reveal a subsidiary role of the hypodermis, which is new. Of particular elegance are RNA-seq studies of individual tissues from worms with tissue-specific daf-2 knockdown.

Overall, I have mixed feelings about this study. First, the technical standard, in terms of use of new methodologies, experimental design and so forth is excellent. I found it fascinating to see these old questions addressed again with new, better tools. On the whole the paper is well written, and I found it an interesting read. In terms of novel insight: while there is nothing startling here, this is an important study in terms of the many questions that it re-answers with greater clarity. However, there are serious problems with the presentation of the work, which I suspect might originate from the authors' anxiety about having the study accepted by a high impact journal. These problems are of two sorts: firstly, recurrent efforts to exaggerate the importance of the findings, including by excessively speculative interpretations of the data. Second, by failing to make clear the extent to which various questions addressed here have also been addressed in previous studies, possibly from worry that this will reduce the perceived impact of the work. In fact, making clear what previous work has been done and how exactly the new work relates to it will increase the value of the study. There are so many studies of aging in *C. elegans* by now that reference to all relevant early studies becomes increasingly important; without this, the field risks degenerating into so many amnesiacs repeating the same experiments over and over.

This is a very good study, which is marred by some rather poor data interpretation (particularly what genuinely useful insights follow from the data), insufficient contextualization, and too much spin.

Major comments

1) The authors concluded in several places, including abstract and introduction that the role of IIS in aging in the intestine implies that a link to dietary restriction has been discovered. This is surely overstated given that no experimental data involving DR is provided. At most, speculating about a possible link warrants brief consideration in the discussion.

2) A stated major conclusion (e.g. emphasized in the title) is that *daf-2* knockdown in the gut extends lifespan without effects on development and reproduction. This seems problematic: why is this important? As expressed, the implication seems to be that this implies that aging could be slowed in humans without side effects, but this seems to me facile sales talk; and besides, it was shown long ago that fertility is irrelevant to aging in *C. elegans*. Some *daf-2* mutants extend lifespan without affecting fertility. The lack of effect on development or fertility seems unimportant to me (the lack of effect on development also argues against a DR effect; DR typically slows development).

3) In the interpretation of the data it is assumed that IIS in the gut regulates aging throughout the worm. But another possibility is that bacterial invasion of the gut limits wild type lifespan. *daf-2* mutant longevity is to some extent attributable to increased resistant to bacterial infection, see Garsin, D.A., et al 2003. Long-lived *C. elegans* *daf-2* mutants are resistant to bacterial pathogens. *Science*. 300, 1921. Podshivalova, K., Kerr, R. and Kenyon, C., 2017. How a mutation that slows aging can also disproportionately extend end-of-life decrepitude. *Cell Reports*. 19, 441–450. This alternative possibility needs to be considered in data interpretation.

I would strongly recommend that this be addressed by comparing lifespans of worms with tissue specific and global knockdown of *daf-2*, plus N2 and *daf-2(e1370)* controls, in the presence and absence of bacterial proliferation (e.g. by using an antibiotic). Digging deeper for new insight in this way could strengthen a study which, though excellent in many ways, is slightly light on real novelty.

Detailed comments

24. insulin/IGF-1

68. “tried to identify the different functions” - surely these studies were all trying to understand where IIS acts to control aging?

“special emphasis” - this is not clear enough. Critical issues here are whether IIS regulates aging from a single tissue, or acts everywhere; and to understand aging by understanding where IIS acts and how.

77. to reliably determine

82. but not in the germline?

83. a tissue-

89-91. This is excessively speculative and looks like spin.

92. lifespan-regulating

93-99. Again, this is horrible spin, and spin must be removed. In former times, equivalent spin might have said that insulin controls metabolism, proving that this pathway acts by controlling mitochondrial ROS. The aging field needs to be better than this. “molecular signature of longevity” is rhetorical

(sounds like more than it is), and inappropriate. What is needed here is a sober account of what exactly the conclusions are that the data supports, particularly the new ones.

101. Better “daf-2 is widely expressed” - no need for the sales talk.

106. jellyfish GFP - you mean the original one or the enhanced, engineered jellyfish GFP?

117. Not indicate, suggest - care to avoid over-interpretation of data

133. Body wall muscles

135. The NuGFP approach is great, but I worry about the possibility of false positive data. Is there evidence that some genes marked with NuGFP do not show expression everywhere? If not, this caveat need to be made clear.

152. expression was detected from the bean stage

155. insulin/IGF

159. knock-in strains (?)

173 onwards. As written this rather overstates the novelty of the findings. In the early 1990s, the conclusions of the Wolkow et al and Libina et al studies clearly conflicted, and it was also clear that the results of the latter study could be more reliably interpreted. The account here needs to explain the conflict between the Wolkow and Libina studies, and the likelihood that conclusions of the latter were correct. Then the new results can be interpreted against that background: they confirm the Libina study conclusions, and create certainty - which is valuable.

180. The interpretation is imprecise/overstated. Yes, the intestine is important; it could be that it regulates the rest of the worm, or is the site of life limiting disease. But, as in the Libina study, there are signs that other tissues may be important too, just not sufficient themselves: Libina shows that daf-16 rescue in the gut only partially restores daf-2 longevity to daf-16; daf-2 worms; and here, AID in all daf-2 expressing tissues extends lifespan much more than AID in the gut alone.

186. FUdR is incorrect: it should be FUDR. The use of FUdR is a meme/error in *C. elegans* papers. Given the choice of repeating a traditional error, or being correct, the right path scientifically is clear.

190. Perhaps this is related to vulval expression of daf-2?

192. Not unequivocally - again, this is overstated in a way that weakens the paper.

192. “This has implications beyond clarifying the purported role” - this is beating about the bush - conclusions needs to be clearly drawn and stated. It argues against it.

193. “which has generated both excitement and uncertainty” - more sales talk, which weakens the paper.

196. Again, the link to DR is speculative, and without direct evidence. More suitable for a brief comment in the discussion only.

197-98. This is gross over-interpretation - delete.

203. Again, the Libina paper work needs to be explained first, and then how the new work adds to that explained.

215. Again, this misses the possibility that intestinal pathology (e.g. infection) is life limiting.

216. Conclusions drawn from data are weak. Surely what is novel here is that it shows that daf-2 action in the intestine itself is controlling DAF-16 localization.

224. As already pointed out (180), these conclusions are surely unsafe.

228. Again, this is overstated.

230. "with previously studied daf-2 mutations" given that daf-2 alleles include daf-2(+)

234. "mild" is subjective - is 100% dauer formation mild?

236. It should be pointed out that daf-2 RNAi does not cause a Daf-c phenotype either. It might be interesting here to test for a Hid phenotype (high temperature Daf-c, 27°C), see Ailion, M. and Thomas, J., 2000. Dauer formation induced by high temperatures in *Caenorhabditis elegans*. *Genetics*. 156, 1047-1067.

251. "the longevity effect was successfully separated from undesirable developmental and reproductive effects" - as discussed above, "successfully" and "undesirable" are relatively meaningless. The original preoccupation with fertility and lifespan, which goes back as far as the Friedman and Johnson age-1 paper had to do with the old disposable soma theory.

257. Introductory sentence is rather vague.

259 onwards. It should be borne in mind that most of these genes where mRNA levels are affected by daf-16 are not directly regulated by the DAF-16 transcription factor. The following arguments fail to take this into account and as a result lead into error.

272. "clean" longevity phenotype - this needs to be rewritten in precise terms. This is a good experiment, though.

282-4. Longevity etc. This is surely not saying anything useful.

295. The use of fib-1 and M28.5/snu-13 needs introduction; what exactly motivated these experiments?

302. Surely the reduction in rRNA is broadly consistent with the reduced levels of protein translation. If I recall correctly, work from the Braeckman lab on proteomics shows reduced levels of ribosomal proteins in daf-2 mutants.

303. Inappropriate conclusions. There is no evidence that the changes in fatty acid and peroxisomal mRNA levels plays any causal role in effects on lifespan.

312. This RNA-seq work with isolated tissues is really good.

321. is not important for IIS effects on lifespan ?

332. "Decreased protein synthesis is also a molecular signature of longevity by Dietary Restriction (DR)" It is misleading to refer to reduced protein synthesis as a signature in this context. Signature implies something highly specific. It is like saying "getting wet is a signature of being rained on".

334-336. This is a bad argument, since most of the effects of daf-16 on gene expression are indirect. Changes in gene expression here do not imply daf-16 action within the cell type tested.

338. Surely it should be mentioned earlier that degradation of daf-2 in the gut causes daf-16 nuclear localization in the hypodermis.

341. Here what is missing is a brief introduction to the topic of FOXO to FOXO signaling, which is discussed in the Libina paper, and see also Alic, N., et al, 2014. Cell-nonautonomous effects of dFOXO/DAF-16 in aging. Cell Rep. 6, 608-16.

358. *gst-30*. It is not clear why this is discussed.

361. The summary here fails to give a clear sense of what conclusions from the present study have been added to those of previous studies.

444. Again, the conclusions need to follow from a measured assessment of what the data newly demonstrates, taking care to avoid rhetorical statements (esp 446-450) and spin.

750. Journal name.

The figures are generally excellent, as is the technical standard of the paper throughout.

Reviewer #3 (Remarks to the Author):

Zhang et al. use CRISPR knock in and auxin induced degradation to revisit a previously confusing set of findings around the tissue specificity of the insulin signaling pathway in longevity in *C. elegans*. Previous work has shown that although this canonical signal transduction pathway functions from the receptor down through the phosphorylation cascade to the transcription factor FOXO/DAF-16, the tissue in which those two players were shown to work were different. In genetic studies suggested that the cascade from InR to FOXO was not as depicted an intracellular cascade but an inter-tissue one. Along with a recent paper by the Ewald lab in eLife, Zhang et al revisit this question, but better equipped to do so with CRISPR than the field was in the 1990s. Using endogenous tagging of DAF-2 and DAF-16 they show beautifully that both are expressed as one might expect ubiquitously. Unlike previous reports or the Ewald paper, Zhang et al.'s data suggest the primary site of action for both components of the IIS is intracellularly in the intestine. This longevity effect can negate some of the pleiotropic effects of the genetic mutants. Via tissue specific isolation and sequencing, transcriptional programs down stream of these perturbations are shown, showing cell autonomous and non cell autonomous effects.

Overall these studies are timely and the experiments are beautifully executed. I do not feel that overlap with the Ewald study impacts the impact of this paper or its value to the field or suitability to Nat Coms. Indeed, these data are in my opinion more compelling and more diligently executed.

- The finding that neuronal DAF-2 degradation doesn't fully or even greatly extend lifespan is in stark contrast with the work by Venz et al. and is also somewhat surprising given previous studies. Some more discussion around this point could be valuable. For example, Venz et al. use the *rab-3* promoter with the *tbb-2* 3'UTR while the authors of this paper use the *rgef-1* promoter with the *unc-54* 3'UTR. Could any differences be attributed to the different promoter or UTR use? Additionally, in line 425, the authors use the mention of FIRKO mice to strengthen their case that IIS regulates longevity from the intestine. However, the authors should not neglect that there have been multiple studies showing brain-specific KO of IIS in mice extends lifespan (Taguchi 2007 and Kappeler 2008) and mentioning these studies might provide readers with a more comprehensive view of the current literature.
- Controls showing the effect of Auxin alone at 1mM on lifespan should be included.
- Authors say that intestinal DAF-2 degradation has no effect on development, but Figure 5B shows a clear developmental delay. Although it is considerably less than the *daf-2(e1370)* mutant, the authors should not say that intestinal loss of DAF-2 "had no adverse effect on development," but rather should emphasize that tissue-specific targeting greatly reduces development impairments, yet does not

entirely eliminate it.

- Figure 6 shd be moved to a supplementary figure. Given the tissue specific data in figure 7 supersede it seems less critical to the work. In addition lifespans I figure 6 have very short and somewhat flat (ie not sigmoidal) curves for daf-2 which suggests that are not optimal.
- In light of the small yet significant effects of the neuronal and other tissue specific lines the conclusion is unequivocal and doesn't match the data, shd be toned down.

Minor Points

- Venz et al. (originally on BioRxiv) has now been published in eLife: [10.7554/eLife.71335](https://doi.org/10.7554/eLife.71335)
 - Explain acronyms upon first use, such as "BWM" in Line 133
 - For Figures S4C-S4J and Figure S5, how long were the animals on auxin before imaging and what age are the animals when they are imaged? Did you ever check in older animals to make sure that auxin-inducible degradation is efficient throughout aging (especially in the neuronal line, to make sure this isn't a reason you don't see lifespan extension upon neuronal degradation)?
 - Better definition of "Longevity regulating pathway-worm" in Figure S7A. Especially if you are going to mention this category in the text, it is worthwhile to elaborate on what genes are within this category.
- * CRCT-1 typo shd be CRTC-1

Reviewer #1 (Remarks to the Author):

*In this study, Zhang et al. exploit the versatility of Crispr editing and the AID technology to bring clarity to an old discussion: which are the tissues relevant to the strong longevity phenotype of the IIS mutant *daf-2(e1370)*? First they show that both DAF-2 and its target DAF-16 are ubiquitously expressed throughout the worm body. Tissue-specific AID degradation of both proteins consistently show that they are required in the intestine to modulate longevity - neurons, hypodermis and germline seem to play a minor role. Intestinal degradation of DAF-2 results in a much narrower range of phenotypes (including longevity) compared to the pleiotropic effects of *daf-2(e1370)* mutation. Transcriptomics confirms that reduced protein and RNA metabolism may be central to worm longevity. Finally, IIS induced changes in the intestine also triggers a transcriptional response in other tissues.*

*This manuscript is very well written and presents a solid data set that includes the necessary checks and controls. Although some parts may seem a mere confirmation of some existing studies (which are appropriately cited by the authors), this study is relevant as it finally untangles some conflicting or hard-to-interpret data from the past. It is a big leap forward in the understanding of the role that tissues play in IIS-induced longevity in *C. elegans*. Therefore, I recommend acceptance of this manuscript after minor revision based on the specific comments below.*

Response:

We are grateful that you appreciate the quality and the value of this work. Also, thank you very much for your constructive comments.

Specific comments

*1) Line 74: The authors make a point that studying the expression pattern of poorly expressed genes or genes with complicated structures is difficult. However, they should mention the existence of alternatives such as cell- or tissue-specific transcriptomics (datasets are publicly available in on-line databases, e.g. Gexplore, CenGen). This work already showed broad expression of *daf-2* and *daf-16* in virtually all tissues.*

Response: Thank you very much for pointing it out! We have added this information to the revised manuscript and the related text is copied below.

Page 3, line 76-85, added and revised:

The main source of the confusion is that the exact expression pattern of *daf-2* is unclear..... Cell- or tissue-specific transcriptomic studies suggest that *daf-2* is widely expressed, but it needs confirmation using orthogonal methods as omics data come with a certain level of false identifications. In contrast to the lack of consensus on *daf-2* expression.....

2) Line 197: At several instances in the manuscript, the authors suggest similar downstream mechanisms of lifespan extension for IIS and dietary restriction – which seem to make sense based on the presented data. However, at first glance, the mechanistic link between IIS and DR may look contradictory to the earlier findings showing that DR-induced lifespan extension is often independent of DAF-16 (nicely summarized in Greer and Brunet, 2009 – Aging Cell). The authors may want to clarify this apparent contradiction in the discussion section (e.g. by stating that DR regimens may bypass intestinal IIS and directly cause nutritional scarcity in the worm body).

Response: This is an important and helpful comment. Reviewer 2 made a similar point. Thank you both! We previously favored a view that reduction of IIS may have induced a weak DR effect, which down-regulates protein synthesis. Prompted by reviewers' comment, we reconsidered and abandoned that view. We have revised the manuscript as follow.

Page 13, line 502-505, added:

IIS and DR are two evolutionarily conserved mechanisms that regulate ageing. Although they are distinct processes, they share commonalities: both sense nutrients and both regulate protein synthesis. This study accentuates these commonalities.....

3) Line 210: Did the authors make a calculation mistake? According to the data in table S1 (and visual confirmation in figure 4G) the intestinal DAF-16-mediated lifespan extension of *daf-2(e1370)* is 'only' 72.2%, not 90.3%. Since this is one of the major findings of the paper, this number is relevant. I calculated this percentage as $100 * (1 - (30.50 - 22.63) / (50.95 - 22.63))$.

Response: Thank you very much for catching this error. We have corrected it in the revised manuscript (line 229).

4) Line 244: This remarkable phenotype needs some discussion. The authors hypothesize that DR and IIS may have similar downstream molecular mechanisms to support longevity (assuming that a DAF-2 deficient intestine may provide the body with less nutrients?). How can this not have any repercussions on fertility? It is well-known that all kinds of DR negatively affect fertility. How can this be reconciled with this fully fertile, yet long-lived, intestinal DAF-2-deficient worm?

Response: This is related to the comment on Line 197. We reflected on the reviewer's comments and revised our conclusion. We previously favored a view that reduction of IIS may have induced a weak DR effect, which down-regulates protein synthesis. We have abandoned that view. Whether or not reduction of IIS induces a weak DR effect cannot be concluded, but it is certain that reduction of IIS down-regulates protein synthesis (this and many other studies), as does DR (from the literature). We thank the review for raising this important point.

5) Line 291-302: This paragraph is not very clear. It is based on survival experiments with a limited number of worms (a single lifespan experiment with less than 100 worms). It looks like all knockdowns have a similar but modest lifespan extending effect which seems to support the claim that interfering with rRNA maturation extends lifespan. However, likely due to limited sample size, this effect is borderline non-significant in one case, and significant in the other two cases. This, despite the similar size effect (13-17%), makes this tiny data set hard to interpret. Moreover, it is not clear to me what the point is to test this in a *daf-2* background. Transcriptomics data suggest ribosome biogenesis is down in *daf-2* and this may (partially) cause longevity. Interfering with rRNA maturation further increases *daf-2* lifespan (as it likely does in wild type to a similar extent). Can you conclude anything specific from that? The hypothetical case where only lifespan extension would be observed in WT but not (further) in *daf-2* would be more suggestive that *daf-2* longevity benefits from reduced rRNA maturation/ribosome synthesis. I am not convinced that this little data set adds to the story.

Response: Another helpful comment! We repeated this lifespan experiment twice with more than 120 worms per RNAi condition. We found that knocking down either *fib-1* or *M28.5* reproducibly and significantly increased WT lifespan by 23.6% or 19%, respectively. We have used this new

lifespan result to replace the original Figure 6D and revised the manuscript accordingly.

Page 8, line 322-327, revised:

.....Further, knocking down either *M28.5/snu-13* or *fib-1*, two RNA metabolism genes that are down-regulated in the *daf-2* mutant, extended WT lifespan by 19.0% or 23.6%, respectively (Fig. 6d). *M28.5/snu-13* encodes a conserved protein functioning in rRNA processing and mRNA splicing, and *fib-1* encodes the *C. elegans* fibrillarlin, which catalyzes 2'-O-methylation of pre-rRNA and U6 snRNA. Lifespan extension by *fib-1* RNAi has been reported.....

6) Line 411: *It is important to note that the entire study is dependent on the absolute tissue-specificity of the TIR1 promoters used. Some leaky expression may have large effects on the resulting phenotypes and final interpretation (e.g. rgef-1 is also expressed in the coelomocytes and the pharyngeal gland, lim-7 is expressed in many cell types, including neurons). Nevertheless, I appreciate the extensive controls in figures S4 and S5, but care should be taken and this should be mentioned.*

Response: We appreciate the reviewer's comment. Except for the data shown in Figure S4 and S5, we also verified the tissue-specificity of each promoter by constructing transgenic NuGFP reporters or mCherry::NLS^{EGL-13} reporters (data added in the new Supplementary Fig. 4c-g). Leaky expression of *rgef-1* and *lim-7* was undetectable. Of course, we cannot completely exclude the possibility of a leaky expression below the detection limit.

We have revised the manuscript as below:

Page 5, line 175-176, added:

.....Before using these five promoters, we verified their tissue specificity by expressing NuGFP or mCherry under their control (Supplementary Fig. 4c-g)

Page 11, line 432-434, added and revised:

.....Although we cannot completely exclude the possibility that leaky expression below the detection limit may result in an overestimation of the contribution of IIS in some tissues, a consensus is forming that intestinal DAF-2 has a substantial effect on lifespan regulation.....

7) Line 435: *in this paragraph, the authors discuss the relevance of their findings and refer to IIS in a context beyond C. elegans. What I missed here is a critical note about the functional diversification of the IIS in mammals, where it has evolved into separate IGF and Ins signalling. A reflection on this (and where to act for possible future anti-ageing treatments) would have been more complete.*

Response: Thank you very much for your comments. We have revised the manuscript as suggested.

Page 12, line 480-483, added:

.....However, it must be kept in mind that IIS in mammals is more complex than in *C. elegans*, as it bifurcates into insulin signaling and IGF-1 signaling. Both IR and IGF-1R are widely expressed in mammals (<https://www.proteinatlas.org> and <http://www.informatics.jax.org>).....

Page 13, line 493-494, revised and added:

.....Insulin signaling in mammals or insulin/IGF-1 signaling in invertebrates, is a master regulator

of glucose metabolism and lipid metabolism.....

Typos and small errors

1) Line 394, 432: *this study has been published now in eLife.*

Response: Thank you! We have updated the citation in the revised manuscript.

2) Figure 3: *For tissue comparison, it would be better to use the same time scale for all panels (0-80 days)*

Response: We tested the visual effect of using the same time scale as the reviewer suggested, but too much white space was left for panels with the maximal lifespan less than 50 days. So, we have kept the original time scale in the revised manuscript for aesthetics.

3) Figure 5D: Legend Y-axis: *ug TAG/mg protein ('I' can be deleted)*

Response: we have deleted “I” from the legend Y-axis.

4) Figure 6 & S7: *Avoid using red for the text (ribosome related pathways), this is confusing as 'red' also means 'upregulated' in this figure.*

Response: Thank you for this suggestion. We have changed the color of the text in the Fig. 6 and Supplementary Fig.12 (original Figure S7A).

5) Figure S4'B: *mRuby (not mRubby, twice) – Pdpy-7 (not Pdyp-7)*

Response: We have corrected these mistakes in the revised manuscript.

6) Figure S8B: *'Events', why is this a range? Is this number of sorted cells over different replicates?*

Response: Yes. We have rephrased it as “Number of NuGFP-positive cells sorted” in the new Supplementary Fig. 13b (original Figure S8B) to avoid such confusion.

Reviewer #2 (Remarks to the Author):

The mechanisms by which insulin/IGF-1 signaling control aging in C. elegans is a topic that biogerontologists have been laboring over since the late 1980s, and the discovery of the long-lived age-1 mutant, such that a great deal is already known about the subject. This study by Zhang et al. uses new experimental tools, particularly CRISPR-Cas9, to address anew a number of old questions about this topic in a way that brings it into better focus. New methods are used to generate better reporter constructs for studying daf-2 and daf-16 expression, yielding a better understanding of daf-2 expression. Tissue specific degron mediated degradation of daf-2 and daf-16 confirm the importance of the intestine as a major site of action in IIS effects on lifespan, and reveal a subsidiary role of the hypodermis, which is new. Of particular elegance are RNA-seq studies of individual tissues from worms with tissue-specific daf-2 knockdown.

Overall, I have mixed feelings about this study. First, the technical standard, in terms of use of new methodologies, experimental design and so forth is excellent. I found it fascinating to see these old questions addressed again with new, better tools. On the whole the paper is well written, and I found it an interesting read. In terms of novel insight: while there is nothing startling here, this is an important study in terms of the many questions that it re-answers with greater clarity. However, there are serious problems with the presentation of the work, which I suspect might originate from

*the authors' anxiety about having the study accepted by a high impact journal. These problems are of two sorts: firstly, recurrent efforts to exaggerate the importance of the findings, including by excessively speculative interpretations of the data. Second, by failing to make clear the extent to which various questions addressed here have also been addressed in previous studies, possibly from worry that this will reduce the perceived impact of the work. In fact, making clear what previous work has been done and how exactly the new work relates to it will increase the value of the study. There are so many studies of aging in *C. elegans* by now that reference to all relevant early studies becomes increasingly important; without this, the field risks degenerating into so many amnesiacs repeating the same experiments over and over.*

This is a very good study, which is marred by some rather poor data interpretation (particularly what genuinely useful insights follow from the data), insufficient contextualization, and too much spin.

Response: We are grateful that you appreciate the quality of this work. Also, thank you very much for careful reading of this manuscript and for providing us with critical but constructive comments. They are detailed, insightful, and very clear. We have addressed all the concerns raised by conducting experiments and revising the text. We believe that the revised manuscript is a much improved one and we want to thank you for your help.

Major comments

1) The authors concluded in several places, including abstract and introduction that the role of IIS in aging in the intestine implies that a link to dietary restriction has been discovered. This is surely overstated given that no experimental data involving DR is provided. At most, speculating about a possible link warrants brief consideration in the discussion.

Response: We thank the review for raising this important point. We reflected on reviewers' comments and revised our conclusion. We previously favored a view that reduction of IIS may have induced a weak DR effect, which down-regulates protein synthesis. We have abandoned that view. Whether or not reduction of IIS induces a weak DR effect cannot be concluded, but it is certain that reduction of IIS down-regulates protein synthesis (this and many other studies), as does dietary restriction (from the literature). We have revised the text accordingly.

On a different but related issue, we do think that this study connects two areas of ageing research—insulin/IGF-1 signaling and dietary restriction, since we find that for lifespan regulation both DAF-2 and DAF-16 act primarily in the tissue/organ that supplies nutrients to the entire body. It is well established that IIS is a nutrient-sensing pathway that plays a key role in glucose homeostasis and lipid metabolism. Previously, lifespan regulation and nutrient sensing/regulation of metabolism are treated as separate functions of IIS. Our finding that IIS acts primarily in the intestine whose primary function is to digest food and absorb nutrients for the entire body implies that these two functions are connected.

*2) A stated major conclusion (e.g. emphasized in the title) is that *daf-2* knockdown in the gut extends lifespan without effects on development and reproduction. This seems problematic: why is this important? As expressed, the implication seems to be that this implies that aging could be slowed in humans without side effects, but this seems to me facile sales talk; and besides, it was shown long ago that fertility is irrelevant to aging in *C. elegans*. Some *daf-2* mutants extend lifespan without affecting fertility. The lack of effect on development or fertility seems unimportant to me (the lack of*

effect on development also argues against a DR effect; DR typically slows development).

Response: We greatly appreciate the reviewer's concern on the DR effect we had suggested. Reflecting on the evidence, we agree that we cannot make this point and have revised the manuscript accordingly. We thank the reviewers for helping us arrive at the precise conclusions. As to the importance of the finding, here's what we think. Firstly, how *daf-2* extends lifespan is a very important question that has motivated many laboratories to join the quest. Searching with the terms "(daf-2) AND (lifespan OR (life span) OR aging)" retrieved 512 articles in PubMed and 10,500 results in Google Scholar. However, the question is still open despite the intense research for more than two decades. Our finding that loss of DAF-2 in the gut extends lifespan with minimal side effects on development and reproduction will help focus research of the longevity phenotype on (1) the direct targets of IIS in the intestine and (2) how the intestine influences other tissues. Secondly, this and another study (Venz R, 2021 eLife) prove the concept that by manipulating IIS, it is possible to achieve longevity without severe adverse effects, at least for *C. elegans*. Certainly, this has implications beyond *C. elegans*—why do we take an interest in *daf-2* and the nematode worm after all?

3) In the interpretation of the data it is assumed that IIS in the gut regulates aging throughout the worm. But another possibility is that bacterial invasion of the gut limits wild type lifespan. daf-2 mutant longevity is to some extent attributable to increased resistant to bacterial infection, see Garsin, D.A., et al 2003. Long-lived C. elegans daf-2 mutants are resistant to bacterial pathogens. Science. 300, 1921. Podshivalova, K., Kerr, R. and Kenyon, C., 2017. How a mutation that slows aging can also disproportionately extend end-of-life decrepitude. Cell Reports. 19, 441–450. This alternative possibility needs to be considered in data interpretation.

I would strongly recommend that this be addressed by comparing lifespans of worms with tissue specific and global knockdown of daf-2, plus N2 and daf-2(e1370) controls, in the presence and absence of bacterial proliferation (e.g. by using an antibiotic). Digging deeper for new insight in this way could strengthen a study which, though excellent in many ways, is slightly light on real novelty.

Response: We appreciate the reviewer's suggestion. We have performed the lifespan experiment as suggested and provided the data in the added Supplementary Fig. 9. Briefly, we found that feeding dead OP50 slightly shortened the lifespan of intestinal DAF-2 AID worms, but their lifespan was still longer than that of control worms fed dead bacteria. It suggests that loss of intestinal DAF-2 extends lifespan not simply by enhancing the innate immunity to protect worms from bacterial infection. We can also find evidence supporting this idea from published studies (PMID 28423308 and PMID 12817143). Therefore, there is an anti-ageing signal transduced from the DAF-2-deficient intestine to other tissues.

We have added these results in the revised manuscript and copied over.

Page 6, line 208-215, added:

.....Consistently, degradation of DAF-2 in the intestine extended the lifespan of C. elegans living on dead bacteria (Supplementary Fig. 9). In fact, worms lacking intestinal DAF-2 lived a similarly long life on either dead or live bacteria (Supplementary Fig. 9). Therefore, longevity by intestinal DAF-2 AID is more than enhancing innate immunity to reduce death by bacterial infection through the intestine, as could be deduced from earlier studies. Taken together, these results suggest the

existence of an anti-ageing signal sent from the DAF-2-deficient intestine to other tissues.

Detailed comments

1) 24. *insulin/IGF-1*

Response: It has been corrected.

2) 68. “*tried to identify the different functions*” - surely these studies were all trying to understand where IIS acts to control aging? “*special emphasis*” - this is not clear enough. Critical issues here are whether IIS regulates aging from a single tissue, or acts everywhere; and to understand aging by understanding where IIS acts and how.

Response: Very helpful suggestions. We have revised the related sentences.

Page 2, line 71-75, revised:

.....have been tried to identify where IIS acts to control ageing, development, or metabolism in *C. elegans*. Special emphasis has been placed on whether IIS regulates ageing from a single tissue or not. Despite the efforts, no consensus has been reached: experimental data have suggested the nervous system, the intestine, or multiple cell lineages as the sites of lifespan regulation by IIS.

3) 77. *to reliably determine*

Response: “reliably” has been added in the revised manuscript. Thank you!

4) 82. *but not in the germline?*

Response: Yes. Although the cell- or tissue-specific transcriptional datasets and the *in situ* hybridization data suggest the presence of *daf-16* mRNA in the germline, its germline expression was not detected by traditional transgene method.

5) 83. *a tissue-*

Response: “a” has been added.

6) 89-91. *This is excessively speculative and looks like spin.*

92. *lifespan-regulating*

93-99. *Again, this is horrible spin, and spin must be removed. In former times, equivalent spin might have said that insulin controls metabolism, proving that this pathway acts by controlling mitochondrial ROS. The aging field needs to be better than this. “molecular signature of longevity” is rhetorical (sounds like more than it is), and inappropriate. What is needed here is a sober account of what exactly the conclusions are that the data supports, particularly the new ones.*

Response: Thanks for the feedback. We have rewritten the last paragraph of the introduction, as shown below.

Page 3, line 88-104, revised:

In this study, using CRISPR/Cas9 genome editing technology and a tissue-specific targeted protein degradation system, we determined the endogenous expression patterns of *daf-2* and *daf-16* and revisited the question about their sites of action in lifespan regulation. We found that DAF-2 and DAF-16 are both expressed ubiquitously in the somatic and reproductive tissues, and that both regulate ageing of the entire body from the intestine. Further, degradation of DAF-2 in the intestine

activates not only DAF-16 in the same tissue, but also DAF-16 in other tissues—at least in the hypodermis if not in neurons. We show that cross-tissue signaling from DAF-2 to DAF-16 contributes to longevity, but not as much as intracellular signaling in the intestine itself. Whole-worm and tissue-specific RNA-seq data suggest that down-regulation of protein synthesis and RNA metabolism is a key molecular change underlying the *daf-2* longevity. Lastly, degrading intestinal DAF-2 nearly doubles *C. elegans* lifespan with little to no effect on development and reproduction. These findings show that two major pleiotropic phenotypes of reduced IIS can be uncoupled from longevity. They also suggest that genetic regulation of ageing by IIS may operate by affecting nutrient supply from the intestine to the entire body, thus inducing metabolic changes at the organismal level. This will be an interesting question for future research.

7) 101. Better “*daf-2* is widely expressed” - no need for the sales talk.

Response: we have changed the subheading as suggested.

8) 106. jellyfish GFP - you mean the original one or the enhanced, engineered jellyfish GFP?

Response: The original GFP.

9) 117. Not indicate, suggest - care to avoid over-interpretation of data

Response: We have revised the text accordingly.

10) 133. Body wall muscles

Response: It has been corrected.

11) 135. The NuGFP approach is great, but I worry about the possibility of false positive data. Is there evidence that some genes marked with NuGFP do not show expression everywhere? If not, this caveat need to be made clear.

Response: We have used NuGFP as a reporter to verify the specificity of each tissue-specific promoter used. The related imaging data have been added to the revised manuscript (Supplementary Fig. 4c-g). Where NuGFP is expressed is determined by the promoter; it is not expressed everywhere.

12) 152. expression was detected from the bean stage

Response: We have changed the text accordingly.

13) 155. insulin/IGF

Response: Corrected as suggested.

14) 159. knock-in strains (?)

Response: Yes. It has been corrected.

15) 173 onwards. As written this rather overstates the novelty of the findings. In the early 1990s, the conclusions of the Wolkow et al and Libina et al studies clearly conflicted, and it was also clear that the results of the latter study could be more reliably interpreted. The account here needs to explain the conflict between the Wolkow and Libina studies, and the likelihood that conclusions of the latter were correct. Then the new results can be interpreted against that background: they confirm the Libina study conclusions, and create certainty - which is valuable.

Response: As the reviewer points out, this is an important issue. It has caused much controversy in

the field. Unfortunately, the confusion did not subside with the publication of the Libina paper. It has intensified recently as two published papers (Uno et al. 2021 and Venz et al. 2021) and one bioRxiv manuscript (Roy et al. 2021) all claim that the nervous system and the intestine are equally important sites of action for DAF-2 or DAF-16, or both, in lifespan regulation. We have revised the related text in the Results and in the Discussion, as shown below.

In the Results, Page 6, line 217-230, revised:

The longevity phenotype of *daf-2(e1370)* is fully dependent on *daf-16*, therefore one obvious question is in which tissue DAF-16 mediates this effect. **Previously, Libina et al. examined three tissues and found that re-introducing *daf-16* from a transgene to the intestine, but not muscles or neurons, partially restored longevity in the *daf-2; daf-16* double mutant background. However, using tissue-specific RNAi, Uno et al. found that both the neuronal and the intestinal *daf-16* activities are important for lifespan extension by reduction of IIS. Here, we examined the role of DAF-16 in seven tissues in mediating the *daf-2* longevity (Fig. 4 and Supplementary Data 1). We found that auxin-induced degradation of DAF-16 in the neurons, germline, or hypodermis shortened the *daf-2(e1370)* lifespan by no more than 15.6% (Fig. 4a-c), while that in BWM, gonadal sheath, or XXX cells had no effect (Fig. 4d-f). In comparison, degradation of intestinal DAF-16 shortened the *daf-2 (e1370)* lifespan by 40.1% (Fig. 4g), which means that intestinal DAF-16 mediated 72.2% of the lifespan extension by *daf-2(e1370)*.....**

In the Discussion, Page 11-12, line 422-465, revised:

Anti-ageing by reducing neuronal IIS has been an attractive idea since expression of a *daf-2* transgene under a neuronal promoter, but not a muscle or intestinal promoter, was found to suppress the longevity phenotypes of the *daf-2* mutant. However, transgenic expression of *daf-16* under an intestinal promoter, but not a neuronal or muscle promoter, extended the lifespan of the *daf-2; daf-16* double mutant. **This is consistent with intestine-specific DAF-16 degradation shortening *daf-2* lifespan. Recently, three studies using either tissue-specific knockout or tissue-specific protein degradation all report that neuronal DAF-2 and intestinal DAF-2 have comparable effects on lifespan, as loss of either extends WT lifespan by 50% or so. Here, we showed that intestine or neuron-specific degradation of DAF-2 extended lifespan by 94.4% or 18.6%, respectively. Although we cannot completely exclude the possibility that leaky expression below the detection limit may result in an overestimation of the contribution of IIS in some tissues, a consensus is forming that intestinal DAF-2 has a substantial effect on lifespan regulation. However, there is no consensus yet regarding neuronal DAF-2.**

To solve this controversy, each of the above experiments must be examined to ensure the absence or reduction of neuronal DAF-2 and the intactness of intestinal DAF-2. None of the above studies showed direct evidence of the latter, and only this study showed direct evidence of the former. **This study is the only one so far that determined the expression pattern of *daf-2*, and the DAF-2::mNeonGreen or *daf-2::degron::mNeonGreen* KI strain should be a useful tool to visualize the effect of *daf-2* knockdown, knock out, or DAF-2 protein degradation in neurons. Pertaining to the intactness of intestinal DAF-2 in our degradation experiments of neuronal DAF-2, although the weak intestinal DAF-2::mNeonGreen signal did not provide a direct readout, we reason that intestinal DAF-2 was not affected when neuronal DAF-2 was degraded based on the following: (1) the *rgef-1* promoter showed good neuron specificity (Supplementary Fig. 4c); (2) neuronal DAF-2**

was verifiably degraded (Supplementary Fig. 5a); (3) this treatment extended lifespan by 18.6%; (4) intestine-specific degradation of DAF-2 extended lifespan by 94.4%; and (5) if the 18.6% lifespan extension had resulted from an unintended loss of intestinal DAF-2, it would further refute the idea of IIS regulating lifespan from neurons. Lastly, suspecting that the conflicting results might be related to the different neuronal promoters used, we compared side by side the *rgef-1* promoter used in this study and the *rab-3* promoter used in two other studies. As shown in Supplementary Fig. 14, *rab-3p::NuGFP* showed leaky expression in the intestine, whereas *rgef-1p::NuGFP* did not. This suggests that the seemingly strong longevity phenotype of neuronal DAF-2 AID is induced partly and unintentionally by intestinal DAF-2 AID, due to leaky expression of TIR-1 under the *rab-3* promoter.

Reflecting on the intestinal versus neuronal IIS effects on lifespan, it is worth noting *daf-2* RNAi induces a robust longevity phenotype. It is well known that *C. elegans* neurons are refractory to RNAi, whereas other cells including the intestinal cells are susceptible. Given what we find in this study, longevity by *daf-2* RNAi can be well explained: the intestine responds readily to *daf-2* RNAi and the intestine happens to be the primary site of action for IIS in lifespan regulation.

16) 180. The interpretation is imprecise/overstated. Yes, the intestine is important; it could be that it regulates the rest of the worm, or is the site of life limiting disease. But, as in the Libina study, there are signs that other tissues may be important too, just not sufficient themselves: Libina shows that daf-16 rescue in the gut only partially restores daf-2 longevity to daf-16; daf-2 worms; and here, AID in all daf-2 expressing tissues extends lifespan much more than AID in the gut alone.

Response: To test whether the longevity induced by loss of intestinal DAF-2 is due to increased resistance to bacterial infection, we performed the lifespan assay as the reviewer suggested, please see our response to your major concern #3 above.

With respect to the role of other tissues in lifespan regulation by IIS, we agree that loss of DAF-2 activity in non-intestinal tissues is required for full lifespan extension. Thank you for pointing out that the original sentence is not a precise one.

We have rewritten this part as shown below.

Page 6, line 203-207, revised:

The above results demonstrate that for IIS-mediated lifespan regulation, although DAF-2 in the intestine plays the most important role, DAF-2 activity in other tissues is required for the longevity phenotype to the fullest extent (whole-body DAF-2 AID 166.5% > intestine 94.4% + neuron 18.6% + hypodermis 13.7% + germline 6.4% + others 0%)......

17) 186. FUdR is incorrect: it should be FUDR. The use of FUdR is a meme/error in C. elegans papers. Given the choice of repeating a traditional error, or being correct, the right path scientifically is clear.

Response: Our apology for this mistake. It has been corrected in the revised manuscript.

18) 190. Perhaps this is related to vulval expression of daf-2?

Response: This is a good point and has been added to the revised manuscript as shown below.

Page 5, line 200-201, added:

.....This Egl phenotype, possibly having to do with the vulval expression of *daf-2*, causes internal hatching of eggs.....

19) 192. *Not unequivocally - again, this is overstated in a way that weakens the paper.*

192. *“This has implications beyond clarifying the purported role” - this is beating about the bush - conclusions needs to be clearly drawn and stated. It argues against it. this is beating about the bush - conclusions needs to be clearly drawn and stated. It argues against it.*

193. *“which has generated both excitement and uncertainty” - more sales talk, which weakens the paper.*

Response: We have deleted these phrases or sentences in the revised manuscript.

20) 196. *Again, the link to DR is speculative, and without direct evidence. More suitable for a brief comment in the discussion only.*

Response: This is an important and helpful comment. Reviewer 1 made a similar point. Thank you both! We previously favored a view that reduction of IIS may have induced a weak DR effect, which down-regulates protein synthesis. Prompted by reviewers' comments, we reconsidered and abandoned that view. Whether or not reduction of IIS induces a weak DR effect cannot be concluded, but it is certain that reduction of IIS down-regulates protein synthesis (this and many other studies), as does DR (from the literature). We have revised the manuscript accordingly.

21) 197-98. *This is gross over-interpretation - delete.*

Response: It has been deleted.

22) 203. *Again, the Libina paper work needs to be explained first, and then how the new work adds to that explained.*

Response: Thank you for the suggestion. We have rewritten this section as suggested.

Page 6, line 218-226, added and revised:

.....Previously, Libina et al. examined three tissues and found that re-introducing *daf-16* from a transgene to the intestine, but not muscles or neurons, partially restored longevity in the *daf-2*; *daf-16* double mutant background. However, using tissue-specific RNAi, Uno et al. found that both the neuronal and the intestinal *daf-16* activities are important for lifespan extension by reduction of IIS. Here, we examined the role of DAF-16 in seven tissues in mediating the *daf-2* longevity (Fig. 4 and Supplementary Data 1). We found that auxin-induced degradation of DAF-16 in the neurons, germline, or hypodermis shortened the *daf-2(e1370)* lifespan by no more than 15.6% (Fig. 4a-c).....

23) 215. *Again, this misses the possibility that intestinal pathology (e.g. infection) is life limiting.*

Response: Please see our response to your major concern #3.

24) 216. *Conclusions drawn from data are weak. Surely what is novel here is that it shows that *daf-2* action in the intestine itself is controlling DAF-16 localization.*

224. *As already pointed out (180), these conclusions are surely unsafe.*

Response: Thanks for these comments. We have revised this paragraph as shown below.

Page 6-7, line 236-244, revised:

Nuclear accumulation is a sign of DAF-16 activation, due to alleviation of the inhibitory phosphorylation by AKT-1/2 on DAF-16 following inactivation of AKT kinase or the upstream kinases AGE-1 and DAF-2. In adult animals, **we found that endogenously expressed DAF-16::GFP from a KI allele (Fig. 2) accumulated** only in the intestine following degradation of DAF-2 in the same tissue (Fig. 4i). Moreover, 83.2% of the extra lifespan gained by degrading intestinal DAF-2 required intestinal DAF-16 (Fig. 4j). Taken together, these results demonstrate that **DAF-2 action in the intestine itself is controlling the localization and activation of DAF-16, and intracellular signaling from DAF-2 to DAF-16 mediates most of the effect of intestinal DAF-2 on lifespan.**

25) 228. *Again, this is overstated.*

Response: It has been changed as “the major site”.

26) 230. *“with previously studied daf-2 mutations” given that daf-2 alleles include daf-2(+)*

Response: Thank you. We have changed the text as suggested.

27) 234. *“mild” is subjective - is 100% dauer formation mild?*

Response: The word “mild” is gone in the revised version.

28) 236. *It should be pointed out that daf-2 RNAi does not cause a Daf-c phenotype either. It might be interesting here to test for a Hid phenotype (high temperature Daf-c, 27°C), see Ailion, M. and Thomas, J., 2000. Dauer formation induced by high temperatures in Caenorhabditis elegans. Genetics. 156, 1047-1067.*

Response: Thanks for the suggestion. We have discussed the dauer phenotype caused by *daf-2* RNAi in the revised manuscript. Also, following the reviewer’s suggestion, we performed the dauer assay at 27 °C. We found that at this high temperature, 99% of intestinal DAF-2 AID worms formed dauers, but the vast majority of them are transient dauers which, in a couple of days, gradually resumed reproductive development.

We have revised the manuscript as shown below.

Page 7, line 253-262, revised and added:

The canonical *daf-2(e1370)* mutant worms form dauers at ~0.5% penetrance at 20 °C, but this increases to 100% at 25 °C, which is the upper temperature limit for standard *C. elegans* culture. In contrast, worms lacking DAF-2 only in the intestine did not form dauers at 25 °C (Fig. 5a). When the culture temperature was increased to 27 °C, they formed transient dauers: >90% remained as dauers for no more than a few days before they resumed reproductive development (Supplementary Fig. 10). This bears comparison with *daf-2* RNAi, which extends *C. elegans* lifespan and causes dauer formation at 45% penetrance at 27 °C but not at all at 25 °C or lower. The similar effects of *daf-2* RNAi and intestinal DAF-2 AID may be explained by the intestine’s sensitivity to RNAi (see discussion).

29) 251. *“the longevity effect was successfully separated from undesirable developmental and reproductive effects” - as discussed above, “successfully” and “undesirable” are relatively meaningless. The original preoccupation with fertility and lifespan, which goes back as far as the Friedman and Johnson age-1 paper had to do with the old disposable soma theory.*

Response: We have changed the statement as shown below.

Page 7, line 277-279, revised:

.....degrading DAF-2 only in the intestine **uncouples** the longevity **phenotype from the developmental and reproductive defects** associated with a general reduction of IIS.

30) 257. Introductory sentence is rather vague.

Response: This sentence is rewritten and shown below.

Page 8, line 283-284, revised:

The many pleiotropic phenotypes of the *daf-2* mutants make it difficult to identify gene expression changes that are associated specifically with longevity.....

31) 259 onwards. It should be borne in mind that most of these genes where mRNA levels are affected by *daf-16* are not directly regulated by the DAF-16 transcription factor. The following arguments fail to take this into account and as a result lead into error.

Response: Thank you very much for pointing it out. We have added this in the revised manuscript as shown below.

Page 8, line 284-288, revised:

.....A study of 75 publicly available microarray datasets has identified 1,663 **genes that** are up-regulated (Class I targets) and 1,733 **genes that** are down-regulated (Class II targets) **in *daf-2(-)* worms compared to the *daf-2(-); daf-16(-)* control. These differentially expressed genes (DEGs) include both direct and indirect targets of DAF-16.....**

32) 272. "clean" longevity phenotype - this needs to be rewritten in precise terms. This is a good experiment, though.

Response: Thank you! We have rewritten this sentence, as shown below.

Page 8, line 300-303, revised:

Since intestine-specific DAF-2 degradation produced a strong longevity phenotype with significantly reduced side effects, we reasoned that it could help separate gene expression changes associated with longevity from those associated with the developmental or reproductive phenotypes of *daf-2* worms.....

33) 282-4. Longevity etc. This is surely not saying anything useful.

Response: It has been deleted in the revised manuscript.

34) 295. The use of *fib-1* and *M28.5/snu-13* needs introduction; what exactly motivated these experiments?

Response: Thanks for the suggestion. It's rewritten and shown below.

Page 8-9, line 322-327, revised:

.....Further, knocking down either *M28.5/snu-13* or *fib-1*, two RNA metabolism genes that are down-regulated in the *daf-2* mutant, extended WT lifespan by 19.0% or 23.6%, respectively (Fig. 6d). *M28.5/snu-13* encodes a conserved protein functioning in rRNA processing and mRNA splicing,

and *fib-1* encodes the *C. elegans* fibrillarin, which catalyzes 2'-O-methylation of pre-rRNA and U6 snRNA. Lifespan extension by *fib-1* RNAi has been reported.....

35) 302. Surely the reduction in rRNA is broadly consistent with the reduced levels of protein translation. If I recall correctly, work from the Braeckman lab on proteomics shows reduced levels of ribosomal proteins in *daf-2* mutants.

Response: Yes. We have cited this work in the revised manuscript (ref. 56).

36) 303. Inappropriate conclusions. There is no evidence that the changes in fatty acid and peroxisomal mRNA levels plays any causal role in effects on lifespan.

Response: The conclusion has been revised as suggested and shown below.

Page 9, line 329-332, revised:

In summary, intestine-specific degradation of DAF-2 helps identify core gene expression changes underlying the longevity phenotype of *daf-2* worms: down-regulation of RNA and protein metabolism. These are shared molecular signatures of longevity following IIS reduction by different means.

37) 312. This RNA-seq work with isolated tissues is really good.

Response: we are glad to know that you like it. We spent a year and a half on this experiment.

38) 321. is not important for IIS effects on lifespan?

Response: We have rewritten this sentence as suggested. Thank you.

Page 9, line 348-349, revised:

.....That only 26 DEGs were found in muscle cells echoes the finding that IIS in muscles is not important for IIS effects on lifespan.

39) 332. "Decreased protein synthesis is also a molecular signature of longevity by Dietary Restriction (DR)" It is misleading to refer to reduced protein synthesis as a signature in this context. Signature implies something highly specific. It is like saying "getting wet is a signature of being rained on".

Response: We have rewritten this sentence as "Of note, Dietary Restriction (DR), which promotes longevity, also decreases protein synthesis" (Page 9, line 360-361).

40) 334-336. This is a bad argument, since most of the effects of *daf-16* on gene expression are indirect. Changes in gene expression here do not imply *daf-16* action within the cell type tested.

338. Surely it should be mentioned earlier that degradation of *daf-2* in the gut causes *daf-16* nuclear localization in the hypodermis.

Response: We agree with the reviewer that gene expression changes of DAF-16 target genes alone cannot be regarded as direct evidence of DAF-16 activation. There is additional evidence. In the revised manuscript, we have rearranged the order by which the different pieces of evidence are presented. Since nuclear localization typically represents DAF-16 activation, we first show nuclear localization of hypodermal DAF-16 induced by the removal of intestinal DAF-2, followed by verification of non-intestinal induction of 4 Class I DAF-16 targets, of which *mtl-1* and *dod-3* are

likely direct DAF-16 targets (ChIP-Seq/qPCR).

Page 9-10, line 362-374, revised:

Importantly, intestinal DAF-2 degradation induced nuclear accumulation of DAF-16::GFP in the hypodermis during the L1-L2 stage (Fig. 7c and Supplementary Fig. 13f), suggesting that hypodermal DAF-16 is activated. Class I DAF-16 targets, with an enrichment of the DAF-16-binding element in their promoter sequences, are more likely to contain direct DAF-16 targets than the Class II targets. So, from the hypodermal and neuronal DEGs (Supplementary Fig. 13d-e), we selected four Class I targets and constructed promoter reporter lines. Indeed, they were induced outside the intestine following degradation of intestinal DAF-2 (Fig. 7c, Supplementary Fig. 13g-h). Among the four selected Class I targets, *dod-3* and *mtl-1* are likely direct targets of DAF-16. We also verified the induction of *gst-30*, another DEG but not a Class I DAF-16 target, outside the intestine (Fig. 7c). In brief, these data indicate that DAF-16 in non-intestinal tissues is activated to some degree in the absence of intestinal DAF-2.

41) 341. *Here what is missing is a brief introduction to the topic of FOXO to FOXO signaling, which is discussed in the Libina paper, and see also Alic, N., et al, 2014. Cell-nonautonomous effects of dFOXO/DAF-16 in aging. Cell Rep. 6, 608-16.*

Response: We appreciate the reviewer's suggestion. The introduction of FOXO-to-FOXO signaling has been added in the revised manuscript as shown below:

Page 10, line 375-378, added:

So far, we have shown that degradation of intestinal DAF-2 activates DAF-16 not only in the intestine (Fig. 4), but also in other tissues (Fig. 7c-d and Supplementary Fig. 13e-h). This is consistent with previously proposed FOXO-to-FOXO signaling. Using targeted protein degradation, we further investigated.....

42) 358. *gst-30. It is not clear why this is discussed.*

Response: *gst-30* is an example to support the idea that other TFs except DAF-16 may be involved in regulating gene expressions upon degrading intestinal DAF-2: *gst-30* is not a DAF-16 target but shows neuronal expression upon degrading intestinal DAF-2. To make this paragraph read more smoothly, we have deleted this sentence in the revised manuscript.

43) 361. *The summary here fails to give a clear sense of what conclusions from the present study have been added to those of previous studies.*

Response: Thanks! We have rewritten this paragraph as shown below.

Page 10, line 396-399, revised:

In summary, the data above demonstrate that the intestine is the primary action site for IIS to regulate lifespan. However, in addition to intracellular signaling from DAF-2 to DAF-16 within the intestine, DAF-16 in other tissues is also activated in response to reduced signaling from DAF-2 in the intestine and contributes to longevity.

44) 444. *Again, the conclusions need to follow from a measured assessment of what the data newly demonstrates, taking care to avoid rhetorical statements (esp 446-450) and spin.*

Response: Thank you for the feedback. We have rewritten this paragraph as shown below:

Page 13, line 502-511, revised:

IIS and DR are two evolutionarily conserved mechanisms that regulate ageing. Although they are distinct processes, they share commonalities: both sense nutrients and both regulate protein synthesis. This study accentuates these commonalities. In the past, research on how IIS regulates ageing paid little attention to nutrient supply or nutrient sensing of various tissues. Our findings here call attention to the most basic function of the intestine, to nutritional supply and metabolism. As discussed above, the intestine is frequently found to be the responsive tissue/organ downstream of many lifespan-extending conditions which are not limited to IIS and DR. We hope that the findings from this study will be helpful to focus the ever-expanding directions of anti-ageing research.

45) 750. Journal name.

Response: Thank you so much for careful reading of this manuscript! We cited the source data (a different paper) in revised manuscript.

The figures are generally excellent, as is the technical standard of the paper throughout.

Response: We are grateful that you appreciate the quality of this study.

Reviewer #3 (Remarks to the Author):

*Zhang et al. use CRISPR knock in and auxin induced degradation to revisit a previously confusing set of findings around the tissue specificity of the insulin signaling pathway in longevity in *C. elegans*. Previous work has shown that although this canonical signal transduction pathway functions from the receptor down through the phosphorylation cascade to the transcription factor FOXO/DAF-16, the tissue in which those two players were shown to work were different. In genetic studies suggested that the cascade from InR to FOXO was not as depicted an intracellular cascade but an inter-tissue one. Along with a recent paper by the Ewald lab in *eLife*, Zhang et al revisit this question, but better equipped to do so with CRISPR than the field was in the 1990s. Using endogenous tagging of DAF-2 and DAF-16 they show beautifully that both are expressed as one might expect ubiquitously. Unlike previous reports or the Ewald paper, Zhang et al.'s data suggest the primary site of action for both components of the IIS is intracellularly in the intestine. This longevity effect can negate some of the pleiotropic effects of the genetic mutants. Via tissue specific isolation and sequencing, transcriptional programs down stream of these perturbations are shown, showing cell autonomous and non cell autonomous effects.*

*Overall these studies are timely and the experiments are beautifully executed. I do not feel that overlap with the Ewald study impacts the impact of this paper or its value to the field or suitability to *Nat Coms*. Indeed, these data are in my opinion more compelling and more diligently executed.*

Response: Thanks for your very positive comments and helpful suggestions. We are pleased to know that you think highly of this work.

*1) The finding that neuronal DAF-2 degradation doesn't fully or even greatly extend lifespan is in stark contrast with the work by Venz et al. and is also somewhat surprising given previous studies. Some more discussion around this point could be valuable. For example, Venz et al. use the *rab-3* promoter with the *tbb-2* 3'UTR while the authors of this paper use the *rgef-1* promoter with the *unc-54* 3'UTR. Could any differences be attributed to the different promoter or UTR use?*

Response: We appreciate the reviewer's suggestion. To examine whether the inconsistency between this study and the Ewald study was due to the different promoter or 3'UTR used, we performed a side-by-side comparison by constructing *NuGFP* reporter strains. Worms expressing *rab-3p::NuGFP::tbb-2 3'UTR* showed leaky expression in the posterior intestine (Supplementary Fig. 14b). In contrast, no leaky expression in the intestine was detected of *rgef-1p::NuGFP::unc-54 3'UTR* (Supplementary Fig. 14a). Therefore, it is very likely that the 50% lifespan extension by degrading neuronal DAF-2 in the Ewald study resulted in part from the unintentional loss of intestinal DAF-2.

Page 11, line 447-448, added:

(1) the *rgef-1* promoter showed good neuron specificity (Supplementary Fig. 4c).....

Page 12, line 452-459, added:

.....Lastly, suspecting that the conflicting results might be related to the different neuronal promoters used, we compared side by side the *rgef-1* promoter used in this study and the *rab-3* promoter used in two other studies. As shown in Supplementary Fig. 14, *rab-3p::NuGFP* showed leaky expression in the intestine, whereas *rgef-1p::NuGFP* did not. This suggests that the seemingly strong longevity phenotype of neuronal DAF-2 AID is induced partly and unintentionally by intestinal DAF-2 AID, due to leaky expression of TIR-1 under the *rab-3* promoter.

2) Additionally, in line 425, the authors use the mention of *FIRKO* mice to strengthen their case that *IIS* regulates longevity from the intestine. However, the authors should not neglect that there have been multiple studies showing brain-specific *KO* of *IIS* in mice extends lifespan (Taguchi 2007 and Kappeler 2008) and mentioning these studies might provide readers with a more comprehensive view of the current literature.

Response: Thank you for the suggestion. We have revised the manuscript accordingly.

Page 12, line 483-487, added and revised:

.....Heterozygous brain-specific knockout of IGF-1R or insulin receptor substrate-2, which transmits signaling from both IR and IGF-1R, has been reported to extend lifespan in mice along with other phenotypes such as growth retardation or insulin resistance. It would be interesting to find out whether intestinal epithelium-specific degradation or knockout of IR extends lifespan of mice.

3) Controls showing the effect of Auxin alone at 1mM on lifespan shd be included.

Response: The lifespan data are provided in Supplementary Fig. 7. For worms treated with either 1 mM auxin or solvent only (0.25% ethanol), their lifespans are comparable.

We have made it clearer in the revised manuscript.

Page 5, line 180-182, added:

.....We also verified that exposure to 1 mM auxin had no effect on the lifespan of WT worms with or without SCI expression of TIR-1 (*ieSi57*) (Supplementary Fig. 7).....

4) Authors say that intestinal DAF-2 degradation has no effect on development, but Figure 5B shows a clear developmental delay. Although it is considerably less than the *daf-2(e1370)* mutant, the authors should not say that intestinal loss of DAF-2 "had no adverse effect on development," but

rather should emphasize that tissue-specific targeting greatly reduces development impairments, yet does not entirely eliminate it.

Response: Sorry for this inaccurate description and thank you for pointing it out. We have changed this statement throughout the manuscript as follows.

Title, revised: Removal of intestinal DAF-2 nearly doubles lifespan but hardly affects development and reproduction of *Caenorhabditis elegans*

Abstract: line 21-23, revised:

.....Remarkably, loss of DAF-2 in the intestine nearly doubles *C. elegans* lifespan and greatly reduces the adverse effects of genetic *daf-2* mutations on development and reproduction.....

Introduction: line 99-100, revised:

.....Lastly, degrading intestinal DAF-2 nearly doubles *C. elegans* lifespan with little to no effect on development and reproduction.....

Results:

Subheading, revised: Intestine-specific degradation of the insulin/IGF-1 receptor extends lifespan with little to no effects on development and reproduction.

Page 7, line 249-252, revised:

.....We found that intestine-specific degradation of DAF-2, which nearly doubled lifespan (Fig. 3g and Fig. 4g) greatly alleviated the development impairment associated with *daf-2* and caused no reproductive defects (Fig. 5).

Discussion, Page 12, line 489-490, revised:

.....depletion of intestinal DAF-2 doubles lifespan with little to no developmental or reproductive defects (this and others).....

5) Figure 6 shd be moved to a supplementary figure. Given the tissue specific data in figure 7 supersede it seems less critical to the work. In addition lifespans I figure 6 have very short and somewhat flat (ie not sigmoidal) curves for daf-2 which suggests that are not optimal.

Response: We appreciate the comments. Having considered this point and a related one (reviewer 1, specific comment #5), we repeated the lifespan assay in the WT background. The original Figure 6D (in *daf-2(e1370)* background) has been replaced by the newly obtained lifespan result (in WT background).

We think that the key messages conveyed by Figure 6 and Figure 7 are different, so we would like to show both as main figures. Figure 6 is about the core gene expression changes associated with the longevity phenotype following IIS reduction by different means (to tease out gene expression changes associated with pleiotropic phenotype. Figure 7 focuses on whether and how loss of intestinal DAF-2 affects other tissues.

6) In light of the small yet significant effects of the neuronal and other tissue specific lines the conclusion is unequivocal and doesn't match the data, shd be toned down.

Response: We agree with the reviewer that loss of DAF-2 in non-intestinal tissues is required for

lifespan extension to the full. We have revised the text accordingly.

Page 6, line 203-207, revised:

The above results demonstrate that for IIS-mediated lifespan regulation, although DAF-2 in the intestine plays the most important role, DAF-2 activity in other tissues is required for the longevity phenotype to the fullest extent (whole-body DAF-2 AID 166.5% > intestine 94.4% + neuron 18.6% + hypodermis 13.7% + germline 6.4% + others 0%)......

Minor Points

1) Venz *et al.* (originally on *BioRxiv*) has now been published in *eLife*: 10.7554/eLife.71335

Response: The citation has been updated.

2) Explain acronyms upon first use, such as “BWM” in Line 133

Response: We went through the manuscript several more times and spelled out all the acronyms at first use.

3) For Figures S4C-S4J and Figure S5, how long were the animals on auxin before imaging and what age are the animals when they are imaged? Did you ever check in older animals to make sure that auxin-inducible degradation is efficient throughout aging (especially in the neuronal line, to make sure this isn't a reason you don't see lifespan extension upon neuronal degradation)?

Response: We apologized that the original description did not provide enough detailed information. Worms at the late L4 stage were transferred on OP50-seeded NGM plates containing 1 mM auxin at 20 °C, and images were taken 24 hrs later (adult day 1). To ensure the effectiveness of DAF-2/DAF-16 degradation, we transferred living worms to freshly prepared auxin-containing plates every week in case the degradation of auxin over time. Moreover, we have confirmed that neuronal DAF-2 could also be effectively degraded in older animals (Supplementary Fig. 8).

We have revised the manuscript as shown below.

Supplementary Fig. 5 (original Figure S4C-S4J), figure legend, revised:

a-h Specifically degrading DAF-2 in the neurons (**a**), intestine (**b**), body wall muscles (**c**), hypodermis (**d**), germline (**e**), gonadal sheath (**f**), XXX cells (**g**), or whole body (**h**) with 1 mM auxin treatment for 24 hours starting from late L4 stage at 20 °C.

Supplementary Fig. 6 (original Figure S5), figure legend, revised:

Specifically degrading DAF-16::GFP::degron in the neurons (**a**), intestine (**b**), body wall muscles (**c**), hypodermis (**d**), germline (**e**), gonadal sheath (**f**), XXX cells (**g**), or whole body (**h**) with 1 mM auxin treatment for 24 hours starting from late L4 stage at 20 °C .

Page 5, line 182-184, added:

.....Additionally, auxin-induced degradation sustained in old worms, as demonstrated using the neuron-specific DAF-2 AID strain (Supplementary Fig. 8).

5) Better definition of “Longevity regulating pathway-worm” in Figure S7A. Especially if you are going to mention this category in the text, it is worthwhile to elaborate on what genes are within this category.

Response: Point taken. Reviewer 2 made a similar comment (detailed comment #33). All considered, we deleted this sentence in the revised manuscript.

6) *CRCT-1 typo shd be CRTC-1*

Response: Thank you very much for catching this error. It has been corrected.

REVIEWER COMMENTS

Reviewer #1 (Remarks to the Author):

The authors have addressed all my concerns appropriately.
This updated manuscript can be accepted for publication.

Reviewer #2 (Remarks to the Author):

The authors have adequately addressed many of my prior concerns but for several key issues not sufficiently. This relates to my previous, serious concerns about spin in the presentation of this work, and insufficient reference to earlier studies, leading to an exaggeration of claims of novelty. I worry that the authors did not grasp the seriousness of my concerns about these two issues. The outstanding problems relate mainly to my first two main points. There are also new issues with respect to the interpretation of the interesting new gentamycin data.

Main points

1) My first main point has not been adequately addressed (see initial review). The point about the likely links between IIS and DR has been made repeatedly in the past, going right back to the original Kumura et al daf-2 cloning paper in 1997 in Science. The importance of the intestine in IIS effects on lifespan have been a recurrent point of emphasis since the Libina et al 2003 Cell paper.

The statement in the abstract: "As the intestine supplies nutrients to the entire body, our findings connect two areas of ageing research—insulin/IGF-1 signaling and dietary restriction—and call to focus anti-ageing research more on nutrient supply to different tissues" reinforces my concerns, expressed previously, about amnesiac science. This also applies to the end of the introduction, and the discussion.

Moreover, to say that the magnitude of the effect on lifespan is "Remarkable" seems hyperbolic, and smacks of spin. Many trials of many different sorts have sometimes observed large increases in lifespan in *C. elegans* with reduced IIS, up to 10-fold. The marked effect on lifespan with knockdown in the gut alone is noteworthy but, surely, not remarkable. The authors' insistence on such spin weakens the presentation of this paper in my view.

"Decreased protein and RNA metabolism are pro-longevity molecular changes": this is a somewhat fuzzy statement, with a whiff of spin about it. It seems to imply that because these are "pro-longevity" changes then the increased lifespan is attributable to them, but the study contains no direct evidence of that. Also what "metabolism" means here is vague... rates of synthesis? Protein and RNAi content?

2) My second main point has not been adequately addressed, though the authors say that it has. The abstract still states, as part of the "remarkable" findings that gut-specific daf-2 knockdown "greatly reduces the adverse effects of genetic daf-2 mutations on development and reproduction". This still falls foul of the issues I pointed out in my initial review. This, arguably, is spin, and amnesiac. This also applies to the end of the introduction, and the discussion.

3) The inclusion of tests in the absence of bacterial proliferation strengthens the study by providing additional, novel information about the how gut specific daf-2 knockdown affects lifespan, and Figure S9 is helpful. However, the conclusions drawn by the authors from these findings are superficial, and have neglected many of the interesting and useful conclusions that their new data offers. Below is an account of what this data is saying, and I would strongly urge the authors to make some reference to it. Moreover, the opening with "Consistently" creates something of a non-sequitur. Past studies support the view that the longevity of daf-2 mutants is attributable partly but not wholly to resistance to bacterial infection. For example, the Podshivalova et al Cell Reports study reported that bacterial killing (gentamycin) increased mean lifespan by 40% in N2 but only 16% in daf-2(e1368),

though oddly it actually decreased lifespan in *daf-2(e1370)* by 10% (Fig S5). Thus the question is not: does *daf-2* knockdown act entirely by preventing bacterial infection? Rather it is: to what extent (if any) does it act by preventing bacterial infection?

The authors' new data shows that on live *E. coli daf-2(e1370)*, or gut-specific or whole-worm *daf-2* knockdown increases mean lifespan substantially more than on dead, consistent with infection resistance being a part of the Age phenotype. A possible improvement here would be inclusion of statistical tests of whether the magnitude of the Age phenotype is different \pm gentamycin, e.g. Cox proportional hazard analysis.

It is also interesting to see that the new data replicates the odd finding of Podshivalova et al that gentamycin decreases lifespan in *daf-2(e1370)* (Kenyon lab data trustworthy as always!), and also here with gut-specific or whole-worm *daf-2* knockdown.

These findings, taken together with the e1368 data from Podshivalova could imply that e1370 and the *daf-2* knockdown result in complete resistance to bacterial infection and partial resistance in e1368.

That would be consistent with Aging Cell (2021) 20: e13324, which found that pharyngeal infection was blocked fully by e1370 but only partially by e1368 (Fig 2b).

The surprising life-shortening effect of gentamycin in some instances is worthy of discussion. It suggests that gentamycin has antagonistic effects on survival: promoting survival by preventing infection, but also shortening lifespan in a way that becomes evidence when worms are fully infection resistant. One possibility is that gentamycin is directly, mildly toxic to *C. elegans* (possible moreso in *daf-2* mutants). Another is that gentamycin is too destructive to the *E. coli*; bear in mind that *C. elegans* grow poorly on autoclaved *E. coli*. Thus, perhaps gentamycin makes *E. coli* nutritionally inadequate. One might be able to avoid these issues by using other means to prevent bacterial proliferation (e.g. other antibiotics, PAF, or UV irradiation) - I offer this point for discussion, not as a recommendation of the necessity of these tests here!

Reviewer #3 (Remarks to the Author):

I think the authors have done a great job alleviating any concerns I had and I recommend publication as is.

REVIEWER COMMENTS

Reviewer #1 (Remarks to the Author):

*The authors have addressed all my concerns appropriately.
This updated manuscript can be accepted for publication.*

Response: We are glad to know that our revision has adequately addressed the questions you had. We thank the reviewer for the insightful comments on the original draft.

Reviewer #2 (Remarks to the Author):

The authors have adequately addressed many of my prior concerns but for several key issues not sufficiently. This relates to my previous, serious concerns about spin in the presentation of this work, and insufficient reference to earlier studies, leading to an exaggeration of claims of novelty. I worry that the authors did not grasp the seriousness of my concerns about these two issues. The outstanding problems relate mainly to my first two main points. There are also new issues with respect to the interpretation of the interesting new gentamycin data.

Response: We thank the reviewer for the patient explanation, which helped us understand the underlying crux of the concerns raised. Also, thank you for the elaborate interpretation of the new lifespan data added to the R1 version.

Main points

*1) My first main point has not been adequately addressed (see initial review). The point about the likely links between IIS and DR has been made repeatedly in the past, going right back to the original Kumura et al *daf-2* cloning paper in 1997 in *Science*. The importance of the intestine in IIS effects on lifespan have been a recurrent point of emphasis since the Libina et al 2003 *Cell* paper.*

1-1. The statement in the abstract: "As the intestine supplies nutrients to the entire body, our findings connect two areas of ageing research—insulin/IGF-1 signaling and dietary restriction—and call to focus anti-ageing research more on nutrient supply to different tissues" reinforces my concerns, expressed previously, about amnesiac science. This also applies to the end of the introduction, and the discussion.

Response: Yes, the commonalities between IIS and DR have been noticed in 1997, when the *daf-2* gene was cloned, and discussed since. Previous studies mainly emphasized the common downstream biological processes targeted by IIS and DR, such as translation and autophagy. This study accentuates what may be in common upstream—altered nutrient supply to various tissues. We have changed this statement in the revised manuscript accordingly.

Regarding the reviewer's comments about the importance of the intestinal IIS in lifespan regulation, Libina et al. arrived at the conclusion in 2003 that DAF-16 acts in the intestine—not neurons—to support *daf-2* longevity, but no consensus has been reached in the field. The conclusion of Libina et al. did not align with that of Wolkow et al., which claimed that *daf-2* acts in neurons—not the intestine—to regulate aging (2000, PMID 11021802). Three recent papers claim that neurons and intestinal cells are equally important for lifespan extension by reduction

of IIS (PMID 34235410, PMID 34505574, and PMID 35808897). This study resolves this controversy and clarifies that the intestine, not the nervous system, is the primary site for IIS in lifespan regulation.

We have revised the manuscript as shown below:

Abstract:

“..... This study clarifies the confusion about where the anti-ageing signal initiates in the *daf-2* mutant and confirms that insulin/IGF-1 signaling regulates lifespan primarily from the intestine, not neurons. As the intestine supplies nutrients to the entire body, it follows that nutrition and metabolic regulation are of fundamental importance for organismal ageing.”

Introduction:

Page 3, line 104-119, revised:

“This study clarifies the confusion regarding the question of which tissue initiates the longevity signal in IIS mutants. We find that a commonly used neuronal promoter has leaky expression in the intestine, which explains why some “neuron-specific” inactivation of DAF-2 extends lifespan substantially. Our finding that both DAF-2 and DAF-16 act primarily in the intestine to regulate lifespan strengthens an early finding that *daf-16* activity is required in the intestine—not neurons—to support *daf-2* longevity. As the primary function of the intestine is to digest food and to absorb nutrients, initiation of the longevity signal in IIS mutants possibly involves alteration in the supplying of nutrients by the intestine to the entire body, with the consequence of inducing organism-wide metabolic changes. In support of this, whole-worm and tissue-specific RNA-seq analyses show that longevity by loss of DAF-2 is associated most prominently with down-regulation of protein and RNA metabolism, including synthesis and degradation of both types of biomolecules. The clarification that it is the intestine that initiates the main longevity signal in the *daf-2* mutant highlights a commonality between lifespan extension by IIS reduction and that by dietary restriction: nutrient supply and nutrient sensing. “

Discussion:

Page 12, line 482-493, added:

“For the above reasons, we believe that this study clarifies the confusion regarding which tissue initiates the anti-aging signal in the long-lived IIS mutants. The study by Libina et al correctly points out that it is the intestine, but the evidence displayed is not strong enough, nor complete, in the following aspects: 1) intestine-specific expression of DAF-16 from extrachromosomal arrays in the *daf-2*; *daf-16* double mutant worms restored longevity about halfway, leaving it possible that there may be another tissue which is also a major site of action of IIS in lifespan regulation; (2) no tissue-specific functional analysis of *daf-2* to either confirm or reject the conclusion of an earlier study that the *daf-2* gene acts in neurons to regulate lifespan; (3) did not address directly what might be the cause of the conflicting results from different studies. We are glad that the Libina study is backed up by this one and we are hopeful that research can move on from this sticky point.....”

Page 14, line 560-570, revised and added:

“It is noticed that IIS and DR share commonalities: both are evolutionarily conserved mechanisms that regulate ageing and both involve nutrient-sensing mechanisms connected to regulation of protein synthesis and autophagy. It is generally agreed upon that DR reduces mTOR signaling. However, whether DR or inhibition of mTOR intersects with IIS has remained unclear, since most forms of DR extend lifespan in a DAF-16-independent manner. Conversely, there is no direct evidence of reduction of mTOR activity in the long-lived IIS mutants. So, lifespan regulation by IIS and DR seem to invoke distinct signal transduction pathways, but certain downstream biological processes such as translation are targeted by both. Our findings in this study call attention to what is likely the common upstream change—altered nutrient supply to various tissues.....”

1-2. Moreover, to say that the magnitude of the effect on lifespan is “Remarkable” seems hyperbolic, and smacks of spin. Many trials of many different sorts have sometimes observed large increases in lifespan in C. elegans with reduced IIS, up to 10-fold. The marked effect on lifespan with knockdown in the gut alone is noteworthy but, surely, not remarkable. The authors’ insistence on such spin weakens the presentation of this paper in my view.

Response: We have replaced the adverb remarkably with “notably” in the revised manuscript.

1-3. “Decreased protein and RNA metabolism are pro-longevity molecular changes”: this is a somewhat fuzzy statement, with a whiff of spin about it. It seems to imply that because these are “pro-longevity” changes then the increased lifespan is attributable to them, but the study contains no direct evidence of that. Also what “metabolism” means here is vague... rates of synthesis? Protein and RNAi content?

Response: Our apology for this imprecise description. In our study, protein metabolism refers to translation and degradation of protein, and RNA metabolism refers to the transcription and degradation of RNA. It’s rephrased in the revised manuscript as shown below.

Abstract:

“.....Longevity by loss of DAF-2 either in the intestine or everywhere in the body is associated most prominently with a decrease of protein and RNA metabolism, in both synthesis and degradation.....”

Introduction:

Page 3, line 113-116, added:

“.....In support of this, whole-worm and tissue-specific RNA-seq analyses show that longevity by loss of DAF-2 is associated most prominently with down-regulation of protein and RNA metabolism, including synthesis and degradation of both types of biomolecules.....”

Results:

Page 8, line 314-316, added:

“.....it is evident that RNA-seq data revealed extensive down-regulation of multiple pathways related to protein and RNA metabolism, in both synthesis and degradation.....”

Page 9, line 322-334, added:

“..... Notably, RNA-seq analysis underscored down-regulation of genes functioning in protein and RNA metabolism, which includes translation, transcription, protein degradation, and RNA degradation.....”

Page 10, line 374-377, added:

“.....GO terms enriched from the down-regulated genes are related predominantly to protein and RNA metabolism including translation, transcription, degradation of protein, and degradation of RNA.....”

2) My second main point has not been adequately addressed, though the authors say that it has. The abstract still states, as part of the “remarkable” findings that gut-specific daf-2 knockdown “greatly reduces the adverse effects of genetic daf-2 mutations on development and reproduction”. This still falls foul of the issues I pointed out in my initial review. This, arguably, is spin, and amnesiac. This also applies to the end of the introduction, and the discussion.

Response: We have reflected deeply on the reviewer's comments. In our view, the key question is if IIS is to be reduced globally, how to reach a perfect level of reduction and what is this perfect level exactly. Over-reduction of IIS causes adverse effects, but under-reduction comprises the longevity benefit. Additionally, and importantly, how to apply what's learnt from *C. elegans* to mammalian models. The genetic alleles and RNAi of *daf-2* or other IIS genes fell short of providing the information needed. The finding that intestine-specific DAF-2 AID nearly doubles *C. elegans* lifespan with greatly reduced adverse effects provides a better alternative.

The revised text is copied below:

Abstract:

“.....Notably, intestine-specific loss of DAF-2 extends *C. elegans* lifespan by 94% while avoids most of the adverse pleiotropic phenotypes associated with genetic mutations of the *daf-2* gene. Specifically, these long-lived worms reproduce normally, develop slightly slower and store more fat than the wild type.....”

Introduction:

Page 3, line 99-103, revised and added:

“.....Furthermore, degradation of intestinal DAF-2 nearly doubles *C. elegans* lifespan with little or no effect on development and reproduction. In other words, two major pleiotropic phenotypes caused by reducing IIS throughout the body can be uncoupled from the longevity phenotype by manipulating IIS in a tissue-specific manner.....”

Discussion:

Page 13-14, line 522-543, added:

“Studies of a complex question such as aging in a simple invertebrate model serve to provide directions to studies of the same question in complex vertebrate animals. In this regard, the *C. elegans* research on lifespan regulation by IIS has been a showcase example. However, earlier findings from *C. elegans* on how to extend lifespan without adverse effects on development

and reproduction fall short of providing specific directions, as explained below. Potentially, one way to getting only the good effect is to look for special mutations on IIS genes, but this has not met with success. Two null alleles of *age-1* can extend *C. elegans* lifespan 9-10 times, which is the most dramatic longevity phenotype of any IIS mutant recorded to date, but those worms are completely sterile, severely retarded in development, and strongly prone to forming dauers. The canonical *daf-2(e1370)* allele, which doubles *C. elegans* lifespan, is so far the best one, but it is not free of pleiotropic phenotypes in development and reproduction (Fig. 5). Another way is through *daf-2* RNAi, but to what extent IIS activity is reduced by *daf-2* RNAi is not known and the extent of lifespan extension by *daf-2* RNAi is variable from experiment to experiment, but never approaching that by *daf-2(e1370)*. With either mutations or RNAi, the problem encountered is similar: it is unknown whether extraordinary longevity at no fitness cost is achievable by reducing IIS globally, and if so, by how much and how to modulate IIS to that perfect level. Based on the findings in mice and the findings in *C. elegans*, as described above, we suggest that intestine-specific removal or reduction of IIS is a potentially promising approach to achieve mammalian longevity with little to no adverse effects on developmental or reproduction. ”

3) *The inclusion of tests in the absence of bacterial proliferation strengthens the study by providing additional, novel information about the how gut specific daf-2 knockdown affects lifespan, and Figure S9 is helpful. However, the conclusions drawn by the authors from these findings are superficial, and have neglected many of the interesting and useful conclusions that their new data offers. Below is an account of what this data is saying, and I would strongly urge the authors to make some reference to it. Moreover, the opening with “Consistently” creates something of a non-sequitur.*

3-1: *Past studies support the view that the longevity of daf-2 mutants is attributable partly but not wholly to resistance to bacterial infection. For example, the Podshivalova et al Cell Reports study reported that bacterial killing (gentamycin) increased mean lifespan by 40% in N2 but only 16% in daf-2(e1368), though oddly it actually decreased lifespan in daf-2(e1370) by 10% (Fig S5). Thus the question is not: does daf-2 knockdown act entirely by preventing bacterial infection? Rather it is: to what extent (if any) does it act by preventing bacterial infection?*

The authors’ new data shows that on live E. coli daf-2(e1370), or gut-specific or whole-worm daf-2 knockdown increases mean lifespan substantially more than on dead, consistent with infection resistance being a part of the Age phenotype. A possible improvement here would be inclusion of statistical tests of whether the magnitude of the Age phenotype is different \pm gentamycin, e.g. Cox proportional hazard analysis.

It is also interesting to see that the new data replicates the odd finding of Podshivalova et al that gentamycin decreases lifespan in daf-2(e1370) (Kenyon lab data trustworthy as always!), and also here with gut-specific or whole-worm daf-2 knockdown.

Response: We greatly appreciate the reviewer’s insightful interpretation of the new lifespan data. They help bring the conclusion to a deeper level. Statistical results of the Cox proportional hazard analysis were added to new supplementary Fig. 9d. We have rewritten this section as suggested by the reviewer, and the related text is copied below.

Abstract:

“.....The almost doubling of lifespan by loss of intestinal DAF-2 results from systemic changes, attributable not only to local changes such as elevated protection against bacterial infection of the intestine but also to activation of DAF-16 in non-intestinal tissues.....”

Results:

Page 6, line 224-234, revised:

“Previous studies have shown that the *daf-2* mutant worms have enhanced innate immunity and are resistant to bacterial infection, which contributes to their longevity. In this study, we found that worms lacking intestinal DAF-2 lived a long life on bacteria killed or not by gentamicin, about 94% longer than WT worms on live bacteria or 28% longer than WT worms on killed bacteria (Supplementary Fig. 9b). Cox proportional hazard regression analysis revealed that gentamicin reduced the hazard ratio by 0.24, while degradation of intestinal DAF-2 by 0.71 (Supplementary Fig. 9d). The reduction of the hazard ratio by degradation of intestinal DAF-2 is comparable to that by *daf-2(e1370)* or by degradation of DAF-2 throughout the body. These results indicate that longevity by intestinal DAF-2 AID is more than enhancing innate immunity to reduce death by bacterial infection through the intestine.”

Discussion:

Page 14, line 544-550, added:

“Like the *daf-2(e1370)* and the whole-body DAF-2 AID worms, the longevity of intestinal DAF-2 AID worms is partly attributable to elevated protection against bacterial infection (Supplementary Fig. 9). Enhanced resistance to bacterial infection of the intestine is most likely a local effect of DAF-2 AID in the intestine, that of the pharynx is not. The part of the lifespan extension that cannot be explained by enhanced innate immunity is more likely than not a systemic effect of a longevity signal sent from the DAF-2-deficient intestine to the entire body.”

3-2: *These findings, taken together with the e1368 data from Podshivalova could imply that e1370 and the daf-2 knockdown result in complete resistance to bacterial infection and partial resistance in e1368. That would be consistent with Aging Cell (2021) 20: e13324, which found that pharyngeal infection was blocked fully by e1370 but only partially by e1368 (Fig 2b).*

Response: It is an interesting point. However, we don't feel that this paper is the right place to discuss the difference between *e1370* and *e1368* on resistance to bacterial infection, for (1) it's not our original data and we did not repeat the experiment on *e1368*; (2) the issue to be discussed is beyond the scope of this study. Discussion of this in a review paper would be good.

3-3: *The surprising life-shortening effect of gentamycin in some instances is worthy of discussion. It suggests that gentamycin has antagonistic effects on survival: promoting survival by preventing infection, but also shortening lifespan in a way that becomes evidence when worms are fully infection resistant. One possibility is that gentamycin is directly, mildly toxic to C. elegans (possible moreso in daf-2 mutants). Another is that gentamycin is too destructive to the E. coli; bear in mind that C. elegans grow poorly on autoclaved E. coli. Thus, perhaps gentamycin makes E. coli nutritionally inadequate. One might be able to avoid these issues by*

*using other means to prevent bacterial proliferation (e.g. other antibiotics, PAF, or UV irradiation)
- I offer this point for discussion, not as a recommendation of the necessity of these tests here!*

Response: Thank you very much for providing these valuable points for discussion. We have added them in the revised manuscript and the related text is copied below. Again, thank you for patiently and critically evaluating this manuscript. Your comments helped us think deep and hard on the questions debated in the field, the value of this study, and how to express as precisely as possible.

Discussion:

Page 14, line 551-559, added:

“Interestingly, the life-shortening effects of gentamicin in *daf-2(e1370)* worms was once again observed in DAF-2 AID worms, suggesting that gentamycin has antagonistic effects on survival: promoting survival by preventing infection, but also shortening lifespan when worms are fully resistant. One possible explanation is that gentamicin itself is mildly toxic to *C. elegans*, perhaps more so in the *daf-2(e1370)* mutant and DAF-2 AID worms. Another possibility is that gentamycin makes the bacteria nutritionally inadequate, as worms grow poorly on autoclaved bacteria. One might be able to avoid these issues by using other means to prevent bacterial proliferation (e.g., other antibiotics, UV irradiation, or paraformaldehyde.)”

Reviewer #3 (Remarks to the Author):

I think the authors have done a great job alleviating any concerns I had and I recommend publication as is.

Response: We appreciate your positive feedback and the suggestions you have made.

We thank all three reviewers for helping us improve the manuscript. We do think that the twice revised version is a much better one than the original draft.

REVIEWERS' COMMENTS

Reviewer #2 (Remarks to the Author):

The authors have responded thoroughly to my earlier concerns, and the new version of the manuscript is looking good.

I have only a couple of last comments which the authors may wish to consider (but no need to come back to me). The deduction that reducing IIS in the mammalian intestine is quite weak, given that the worm intestine also serves as the equivalent to mammalian adipose and liver; here it is perhaps significant that fat-specific knockout of the insulin receptor can increase murine lifespan (Science. 2003 299 572-4.).

A more minor point: to speak of reduced IIS having no fitness cost is somewhat misleading. One assumes that wild-type IIS levels are optimal in terms of the decision to enter dauer; reduced IIS that increases dauer formation will reduce reproductive fitness. More precise to say that lifespan is increased without any detrimental effects on adult function and health.

REVIEWERS' COMMENTS

Reviewer #2 (Remarks to the Author):

The authors have responded thoroughly to my earlier concerns, and the new version of the manuscript is looking good.

Response #1: We are glad to know that our revision has adequately addressed yours concerns. We would like to express our gratitude one more time for your critical reading of this manuscript. Your detailed and constructive comments guided us to reflect on a few issues and to articulate our thoughts with precision.

I have only a couple of last comments which the authors may wish to consider (but no need to come back to me). The deduction that reducing IIS in the mammalian intestine is quite weak, given that the worm intestine also serves as the equivalent to mammalian adipose and liver; here it is perhaps significant that fat-specific knockout of the insulin receptor can increase murine lifespan (Science. 2003 299 572-4.).

Response #2: We appreciate the reviewer's comment. The related text is as follows (in the discussion). "Lifespan extension by activating intestinal DAF-16 is not unique to *C. elegans*. The worm intestine, in addition to being a digestive tract, doubles as an adipose tissue. In mice, adipose tissue-specific knockout of IR (FIRKO mice) extends lifespan¹¹, and intestinal epithelium-specific knockout of IR alleviates insulin resistance in aged animals⁹². Both treatments activate the mammalian counterpart of DAF-16, the FoxO TFs, in the IR cells. However, it must be kept in mind that IIS in mammals is more complex than in *C. elegans*, as it bifurcates into insulin signaling and IGF-1 signaling. Both IR and IGF-1R are widely expressed in mammals (<https://www.proteinatlas.org> and <http://www.informatics.jax.org>). Heterozygous brain-specific knockout of IGF-1R or insulin receptor substrate-2, which transmits signaling from both IR and IGF-1R, has been reported to extend lifespan in mice along with other phenotypes such as growth retardation or insulin resistance^{93,94}. It would be interesting to find out whether intestinal epithelium-specific degradation or knockout of IR extends lifespan of mice."

Examining the data reported in the literature and those in this study, we find that neither the FIRKO phenotype nor the phenotype of liver-specific IR knockout mice mirrors that of intestine-specific IIS reduction in *C. elegans*. Liver-specific IR knockout mice are diabetic (PMID: 10949030). Although the FIRKO mice are longer lived than control by 18%, their fat mass is reduced by 50-70% and their gene expression profile features most prominently up-regulation of mitochondrial oxidative metabolism¹¹. The intestinal DAF-2 AID worms are long-lived (Fig. 4g-h), their fat storage more than doubled (Fig. 5d), and their gene expression profile features most prominently down-regulation of RNA and protein metabolism (Fig. 6c). Intestinal epithelium-specific knockout of IR alleviates insulin resistance in aged mice, but the effect on

lifespan is not known⁹². So, we wonder whether it extends the lifespan of mice, and if yes, whether it transcriptionally down-regulates RNA and protein metabolism.

A more minor point: to speak of reduced IIS having no fitness cost is somewhat misleading. One assumes that wild-type IIS levels are optimal in terms of the decision to enter dauer; reduced IIS that increases dauer formation will reduce reproductive fitness. More precise to say that lifespan is increased without any detrimental effects on adult function and health.

Response #3: After careful consideration of the reviewer's comment, we think that the reviewer misunderstood our statement. The related text (in the discussion) is: "With either mutations or RNAi, the problem encountered is similar: **it is unknown whether extraordinary longevity at no fitness cost is achievable by reducing IIS globally**, and if so, by how much and how to modulate IIS to that perfect level. Based on the findings in mice and the findings in *C. elegans*, as described above, we suggest that intestine-specific removal or reduction of IIS is **a potentially promising approach to achieve mammalian longevity with little to no adverse effects on developmental or reproduction.**"

The question we are asking is whether it is possible and how to tune IIS to a perfect level that extends lifespan without adverse effects on development and reproduction. Countering the reviewer's point, the *age-1(hx546)* mutant *C. elegans* is long-lived but does not form dauer spontaneously (Dorman JB, Genetics 1995), which shows that it is possible to have the longevity phenotype without the Daf-c phenotype. Therefore, the fitness cost due to increased dauer formation is avoidable.